

# High-resolution modeling of tsunami run-up flooding: A case study of flooding in Kamaishi City, Japan, induced by the 2011 Tohoku Tsunami

Ryosuke Akoh[1], Tadaharu Ishikawa[2], Takashi Kojima[3], Mahito Tomaru[3], Shiro Maeno[1]

[1]Graduate School of Environmental and Life Science, Okayama University, Okayama, 700-8530, Japan
[2]A Professor emeritus, Tokyo Institute of Technology, Kanagawa, 226-8503, Japan
[3]TOKEN C. E. E. Consultants Co., Ltd., Tokyo, 174-0004, Japan

*Correspondence to*: Ryosuke Akoh (akoh@okayama-u.ac.jp)

**Abstract.**

Run-up processes of 2011 Tohoku Tsunami into the city of Kamaishi, Japan, were simulated numerically using 2D shallow equations with a new treatment of building footprints. The model imposes the internal hydraulic condition of permeable/impermeable walls at the building footprint outline on unstructured triangular meshes. Digital data of the building footprint approximated by polygons were overlaid on a 1.0 m resolution terrain model. The hydraulic boundary conditions were ascertained by conventional tsunami propagation calculation from the seismic center to nearshore areas. Run-up flow

calculations were conducted under the same hydraulic conditions for several cases with different building permeabilities. Comparison of computation results with field data suggests that the case with a small amount of wall permeability gives better agreement than the case of impermeable condition. Spatial mapping of an indicator for run-up flow intensity ($Z = U_{max} \times H_{max}$) shows fairly good correlation with the distribution of houses destroyed by flooding. Results of numerical experiments show that concrete buildings arrayed alternately in two lines can prevent seawater from flowing straight to the city center while

maintaining access to the sea. The $Z$ value was significantly lower on streets where many houses were destroyed by the 2011 Tohoku Tsunami.

Key word: Tsunami hazard, numerical modeling, urban area, building footprint, internal boundary condition

## 1 Introduction

Recent urbanization of low-lying coastal areas has increased the potential for property damage, human injury, and death caused by tsunamis. Visual data obtained during the tsunami run-up revealed that arrays of structures in urban areas induced large wave deformation and swift currents on streets, and that the currents washed objects such as garbage, cars, and debris from damaged structures, causing even more damage than tsunami run-up over uniform ground. Prediction of swift currents in urban areas by numerical flow simulation is expected to be important for evacuation programs and for city layout planning

measures to mitigate tsunami damage.

Tsunami simulation models for forecasting wave propagation and deformation from the seismic center to the coast have been developed and improved for decades. These models for high-speed calculations in a wide water body are often based on a set of shallow-water equations on a structured rectangular grid system (Imamura 1995). Models with a rectangular grid system were extended to calculate the tsunami run-up on land by formulating the wavefront propagation on a dry bed (TiTov

et al. 1995, 1998; Synolakis et al. 2008). However, the tsunami run-up simulation described above requires more precise flow modeling by introducing the hydraulic effects of building arrangement.

Building array treatments in urban flood inundation models are classifiable into four types (Schubert et al. 2008; Schubert et al. 2012): building-resistance models (BR), in which large surface roughness is assigned to cells that fall within a building footprint (Liang et al. 2007) or developed parcels (Gallegos et al. 2009; Gallinen et al. 2011); building-block models (BB), in



which spatially distributed ground elevation data are raised to roof-top height (Brown et al. 2007; Hunter et al. 2008; Schubert et al. 2008; Lee et al. 2016); building–hole models (BH), in which building footprints are excluded from the flow calculation area with a free-slip wall boundary condition (Aronica et al. 1998; Aronica et al. 2005; Schubert et al. 2008; Tsubaki et al. 2010); and building-porosity models (BP), in which the impact of buildings in a street block is expressed approximately by

porosity and a drag coefficient in a street block (Guinot 2012; Sanders et al. 2008; Soares-Frazão et al. 2008).

Schubert et al. (2012) compared the performance of the four models described above to case studies of urban dam-break flooding in Baldwin Hills, California, using shallow water equations as governing equations. They concluded that the BR model does not provide an accurate velocity field, but its execution is fast. The BB model demands a fine grid around buildings, which raises the computational cost. The BH model requires simplification of building geometries for reasonably fast execution.

The BP model can best balance accuracy and run-time efficiency, but it has difficulty determining the porosity and drag coefficient from building geometry data.

The BR model is commonly adopted for tsunami run-up simulations (Gayer et al. 2010; Kaiser 2011; Suppasri et al. 2011; Bricker et al. 2015), although the model developers did not predict the velocity field. The water depth distribution is adjustable to the measured data by changing the roughness parameter distribution.

Komatsu et al. (2010) applied the BB model to an unstructured triangular grid system for the city of Kota Banda Aceh of Indonesia to simulate tsunami flooding caused by the 2004 off the Indian Coast of Sumatra Island Earthquake. Conde et al. (2013) applied the same kind of model to simulate flooding in two cities of Portugal caused by the 1755 Lisbon Tsunami. Imai et al. (2013) proposed a combined model in which buildings wider or smaller than the calculation mesh size were expressed respectively with the BB model and the BR model. Then they applied the combined model to a historical tsunami run-up in

Kochi city in 1707.

Liu et al. (2001) assessed the city layout effect on tsunami run-up calculation by application of the BH model with two grid sizes to inundation in a city caused by the 1896 Sanriku Earthquake Tsunami. Akoh et al. (2014) proposed a permeable wall model equivalent to the BH model when the permeability constant was zero. They applied the model to a simulation of inundation occurring in Kamaishi city during the 2011 off the Pacific Coast of Tohoku Earthquake (2011 Tohoku Tsunami

hereinafter). No report of the relevant literature has described application of the BP model to tsunami run-up simulation, probably because it is not easy to identify the values of porosity and building drag coefficient for respective street blocks.

The actual pressure deviation from the hydrostatic assumption in the shallow-flow model might cause some errors, especially near the wavefront and near building corners. Therefore, Phuc et al. (2013) and Fujimoto et al. (2013) respectively used the VOF method (Hirt et al. 1981) and the ISPH method (Aly et al. 2011; Asai et al. 2012) to calculate the 3-D non-

hydrostatic flow field of tsunami run-up in Onagawa City induced by the 2011 Tohoku Tsunami. However, because of the large computational cost, these methods were not sufficiently practical.

For this study, the permeable wall model based on shallow flow equations, which was proposed by Akoh et al. (2014), was used to investigate tsunami run-up details in the Kamaishi city induced by 2011 Tohoku Tsunami using more field data than used in the earlier study. Numerical simulations were conducted for several values of the permeability constant (from 0 to 1),

among which the case with $C=0$ was equivalent to the BH model. Two building array conditions were considered: those before the tsunami arrival and those after many wooden houses in city center had been destroyed by the tsunami waves. An indicator for the run-up flow intensity was also introduced: $Z = U_{max} \times H_{max}$, where $U_{max}$ and $H_{max}$ respectively denote the maximum flow velocity and maximum water depth at each point during the flood. Spatial mapping of the calculated $Z$ value was compared with the distribution of wooden houses destroyed by tsunami waves.




## 2 Site description

Kamaishi City is located at the inner part of the Kamaishi Bay in the southern Sanriku saw-tooth coast of Tohoku District, Japan (**Fig. 1**). The distance between the bay mouth and the seismic center of the 2011 Tohoku Earthquake is only 115 km. A GPS wave gauge placed 20 km offshore from the bay mouth recorded the time series of the water surface displacement induced
by the 2011 Tohoku Tsunami.

**Figure 2** presents the bathymetry and surrounding topography of Kamaishi Bay, where Tokyo Peil (T.P.) +m denotes the elevation in meters above the average sea level in Tokyo Bay, which is the standard elevation unit in Japan. A breakwater to prevent tsunami wave intrusion was built at the bay mouth in 2009 after several tsunami disasters occurred in the 19th and 20th centuries. Nevertheless, the upper part of the structure was destroyed by the first wave of the 2011 Tohoku Tsunami. The
center of Kamaishi City is located on narrow lowland surrounded by mountains inside the bay. The Kamaishi City population of approximately 39,000 is mainly reliant on marine product industries.

**Figure 3** depicts the building distribution in the city center before the earthquake provided by Geospatial Information Authority of Japan (GIAJ), where the colors show materials used in the construction of individual buildings. Approximately 2,500 small buildings were clustered close together in a narrow area. More than half of these buildings were mortared wooden
houses (shown as red). The old coastline was at the southern margin of this dense building cluster. The open space between the old coastline and the present coastline is reclaimed land used as a fishing port, a market, and a loading yard. Most of the steel-frame buildings (shown as yellow) were workshops and storehouses used for marine industries. Concrete panels covered the side faces of these buildings. The black line along the coast represents a concrete seawall, the crown elevation of which was T.P. +4 m.

The height of the first wave of the 2011 Tohoku Tsunami was approximately ten meters at the coast near the city center. A large volume of seawater overtopped the seawall and struck the buildings. Black cells in **Fig. 4** show the buildings destroyed completely (washed away) by the tsunami waves. Gray cells show remaining buildings that were nonetheless severely damaged (GIAJ). Most buildings in the city were damaged severely, among which the destroyed buildings were mortared wooden houses, which are common throughout Japan.


## 3 Methods and materials

### 3.1. Numerical Model

Two-dimensional shallow water equations were adopted for numerical simulations: a continuity equation for an incompressible fluid and momentum equations used under the assumption of hydrostatic pressure without horizontal diffusion terms in Cartesian coordinates. The Godunov-type finite volume method (Godunov et al. 1959) was used to solve the
hyperbolic differential equations. The spatial domain of integration was covered by a set of unstructured triangular cells, which are not necessarily aligned with the coordinates. Therefore, the topography and building footprints were expressed flexibly. The cell-averaged values for water depth, velocity components, and ground surface elevation were assigned at the centroid of each triangle.

By integrating the equation over each triangular cell and by application of Gauss's theorem to the flux integral, a finite-
differential equation for time evolution of variables was obtained. The method of characteristics was applied to the flux terms. Roe's approximate Riemann solver (Roe 1981) was adopted, based on the first-order upwind approach. In the finite differentiation of the momentum source term induced by ground slope, an upwind approach was also adopted to satisfy the C-property condition for avoiding nonphysical oscillations by ensuring the balance with the flux term in the steady condition
(Zhou et al. 2001). The momentum source term induced by bed friction, which was expressed by Manning's roughness, was given by the spatial average.





To model wave front motion, the Eulerian method proposed by Brufau (2002) was adopted to avoid the so-called C-property collapse at the border between a wet cell and a dry bed cell. This method temporarily sets the ground elevation of the dry bed cells adjacent to the wet area as equal to the water surface level in the neighboring wet cell. A more detailed description of the method was presented in an earlier report (Akoh 2014).

**3.2 Assumptions of internal hydraulic condition**

Effects of seawalls on a flood flow are expressed by imposing the following internal hydraulic conditions on line segments where the seawalls are located:

$$q = \begin{cases} 0.35\,h_1\sqrt{2gh_1} & if\ h_2/h_1 < 2/3 \\ 0.91\,h_2\sqrt{2g(h_1-h_2)} & otherwise \end{cases} \quad (1)$$

For those expressions, $q$ is the volumetric flow rate over a unit length of seawall. Also, $h_1$ and $h_2$ respectively denote the water depths above the crown at the upstream and downstream sides (see **Fig. 5(a)**). The first equation and the second equation

respectively show the free overflow and submerged overflow (Honma 1940).

The ordinary BH models exclude building footprints from the calculation area using free slip interior boundary condition. In this study, effects of buildings on a flood flow are expressed by imposing the following internal hydraulic condition on the line segments of building footprint outlines:

$$q = \pm C\sqrt{2g\,|h_{out} - h_{in}|}\,(h_{out} + h_{in})/2, \quad (2)$$

where $q$ denotes the flow rate across a unit length of the wall, $h_{out}$ and $h_{in}$ respectively represent the water depths immediately

outside and inside the wall, $(h_{out}+h_{in})/2$ denotes the average wetted height of the wall, $(h_{out} - h_{in})$ represents the water surface difference across the wall (see Fig**. 5(b)**), $g$ stands for gravitational acceleration, and $C$ is a constant representing the wall permeability resulting from openings such as doors, windows, or cracks and slits caused by wave impacts. Positive and negative signs respectively denote the cases in which $h_{out} > h_{in}$ and $h_{out} < h_{in.}$ In an impermeable condition ($C$=0), the model is equivalent to the BH model.

**3.3 Data sources for mesh generation**

**3.3.1. Ground surface elevation**

Based on GPS elevation monitoring by GIAJ, a large ground displacement occurred within a short time immediately after the first shock of the earthquake. The movement ceased before the first tsunami wave arrival at Kamaishi Bay (GIAJ). **Figure**

**6** shows ground elevation data near the coastline of Kamaishi Bay obtained before and after the earthquake, which indicate approximately uniform subsidence of one meter.

Therefore, the ground elevation data for the numerical flow simulation was referred from a 1.0 m resolution digital elevation model provided by GIAJ based on aerial laser scanning after the earthquake. Because no bathymetry measurements were available for Kamaishi Bay after the earthquake at the time of the present study, the seafloor elevation was estimated by

subtracting one meter from measurements taken before the earthquake (Japan Oceanographic Data Center).

**3.3.2. Seawalls and building footprint*s***

The Digital Base Map for Reconstruction Planning (2011) provided by GIAJ includes a dataset of structure plane figures before the earthquake. The seawall positions were approximated by line segments using GIS software SIS.

The digital base map also includes building footprint outlines as corner positions of polygon geometry. However, overly fine expression of irregular building shapes and tight spacing are expected to degrade the computational efficiency. For that reason, building footprints were simplified: Polygon sides shorter than 2.5 m were eliminated by erasing some corners.



Building gaps narrower than 1.5 m were avoided by aggregating polygons. An example of simplification is portrayed in **Fig. 7**. About 2,742 buildings were found in the digital base map for detailed calculations in the city center area. The number was reduced to about 1,800 after polygon aggregation.

### 3.3.3. Mesh generation

The red line in **Fig. 8(a)** presents the total area for tsunami run-up simulation. The blue dotted line shows the area of detailed calculations in which building footprints were counted. **Figure 8(b)** and **Fig. 8(c)** portray enlarged images of the two areas. For the detailed calculation area, a triangular mesh system was constructed from the position dataset of seawalls and building corners using software (ANSYS® ICEM CFD™), under a minimum angle constraint of 30 deg to avoid computational instability caused by acute angles. The ground elevation at each triangle centroid was obtained by interpolating the 1.0 m resolution digital model provided by GIAJ. Manning's roughness coefficient was assumed as $n$=0.02 because almost all land surfaces in the detailed calculation area were roads and bare ground (except for building footprints).

For the areas of suburbs and water (outside the blue dotted line), triangular mesh systems were constructed using ANSYS® ICEM CFD™ based on the coast locations, bed elevation contour lines, and land-use classification boundaries. Manning's roughness coefficient was assigned as described by Bricker et al. (2015) for each land-use classification, as presented in **Table 1** and as shown with colors in **Fig. 8(c)**. In all, 146,665 computation grid areas were made for the whole calculation area.

### 3.4 Calculation conditions

The hydraulic conditions at the east open boundary of calculation area were given by the conventional tsunami propagation model in the ocean (TUNAMI-N2, Imamura 1996) with rectangular grids. For calculation efficiency, a seven-step, one-way nesting method was adopted with grid sizes from 3,240 m to 10 m. The time increment of computation was ascertained from the CFL criterion for each nesting process. The initial distribution of the water surface setup was obtained from the estimation presented by the Central Disaster Prevention Council (2012). The calculation results were compared with data obtained using a GPS water gauge located 20 km off the mouth of Kamaishi Bay (see Fig. 1). **Figure 9** shows the calculated and measured results for the sea surface displacement at a GPS wave gauge station. The agreement was sufficient. The calculated time series of water surface elevation and momentum flux at the east open boundary were imposed to the tsunami run-up model described in the previous sections.

Numerical simulations were conducted with time increment $\Delta t = 0.025$ s for the cases presented in **Table 2**. In Case-1(a), building walls were assumed to be impermeable ($C = 0$) for building arrays in the city center area before the tsunami run-up, but in Case-1(b), $C = 0$ for building arrays without houses destroyed by tsunami flooding. Case-2(a) – Case-5(a) and Case-2(b) – Case-5(b) were counterparts of Case-1(a) and Case-1(b), in which $C$ was changed respectively as $10^{-3}$, $10^{-2}$, $10^{-1}$, and $10^0$.

### 3.5 Data sources for model verification

### 3.5.1 Tsunami wave height near the coast

Digital photographs taken by residents were analyzed to estimate the tsunami run-up near the coast. **Figure 10(a)** portrays the shooting point and the view angle, shown respectively by the yellow dot and the blue lines, in an area (shown as red) where some concrete buildings withstood the tsunami. The water surface elevation at each time was estimated by comparison with the window height, as measured by the authors after the area was opened for tsunami damage investigation. **Figure 10(b)** presents an example in which the red numbers denote heights from the ground of the lower window frames. The blue numbers



and the black numbers respectively denote the vertical angle differences and the elevation differences of the lower window frame and the water surface from the upper window frame, as measured from the digital image. Based on this analysis, the water surface elevation at the moment was estimated as 6.865 m from the ground.

### 3.5.2. Local highest water surface in the city

An academic joint research group was organized to conduct an extensive survey of the disaster caused by the 2011 Tohoku Tsunami. Their survey covered almost the entire Pacific Coast damaged by 2011 Tohoku Tsunami, as marked by red in **Fig. 1(a)**. For the Kamaishi area, after they had estimated the highest water surface level at several points from water surface traces remaining on poles, roofs, building walls, and the ground, they made the dataset available to other research groups via the internet. We present the data on the map, as displayed in **Fig. 11**.

### 3.5.3. Wavefront propagation on streets

A local resident took a video recording of tsunami waves from the point shown as the yellow dot in the direction indicated by the blue arrow presented in **Fig. 12(a)**. After we selected three images in which the tsunami front had just passed the street through three intersections, *R1*, *R2*, and *R3*, marked by yellow dots in **Fig. 12(b)**, we ascertained the time differences among the images, as shown at the bottoms of **Figs. 13(a)–13(c)**.

## 4 Results

### 4.1 Tsunami wave height near the coast

The colored dots depicted in **Fig. 14(a)** show the respective water surface levels ascertained from the photographic analysis described for **Fig. 10** for four points located near the coast. They are shown with the same symbols in **Fig. 14(b)**. The colored solid lines represent the calculated time series of the water surface level at the location of *P3* (see **Fig. 14(b)**) for four cases. All calculation results were mutually similar. They agreed fairly well with the observations. This outcome suggests that building wall permeability and assumptions of building arrays, with or without destroyed buildings, did not strongly influence the tsunami wave height near the coast.

The calculated highest water surface level was from T.P. 10.2 m (Case-3(a)) to 10.5 m (others) at around 2,320 s after the first earthquake shock, although the actual measured highest water surface level was approximately T.P. 9.8 m at around 2,222 s after the first shock. A possible reason for this difference is that the breakwater at the bay mouth was ignored in tsunami propagation calculations for the ocean because the breakwater collapsed at an early stage of the tsunami event, but the destruction of the breakwater might have dissipated the energy of intruding tsunami waves.

### 4.2 Local highest water surface in the city

**Figure 15** shows the correlation between the measured and calculated highest water surface levels for four cases with symbols used also in **Fig. 11**. Lines show perfect agreement (– –), best fit regression line (——) and a regression line with 1:1 slope (– –) inserted in **Fig. 15**. In the cases of $C = 0.0$ (Case-1(a) and Case-1(b)), where water storage in buildings was not considered, large deviations are found in the western run-up data (×) at the end of inundation region, suggesting that the flow concentrated only on streets in the calculation caused overly efficient flood propagation.



Cases with $C = 0.01$ (Case-3(a) and Case-3(b)) present results better than those obtained for the cases with $C = 0.0$, which suggests that modification of the BH model by considering water storage in buildings might better fit Japanese wooden houses. Calculations for the building arrays after the tsunami impact (Case-(b) series) were less accurate than for those before tsunami impact (Case-(a) series), probably because the highest water surface was generated during the first wave run-up, when houses

had not been destroyed completely.

### 4.3 Wavefront propagation on streets

**Figure 16** depicts snapshots of the inundation depth for Case-3(a) at the three moments when the calculated tsunami wavefront reached the three intersections denoted by ***R1***, ***R2***, and ***R3*** (see **Fig. 12(b)**), where the tsunami front passage was

captured using a digital video camera by a resident. The red numbers below **Fig**. **16** represent the time passage between the tsunami front impacts. Comparison of the measured results shown at the bottom of **Fig. 13** reveals that the calculated time differences agree well with the observed time differences.

## 5 Discussion

### 5.1 Permeability constant

$C$ is a parameter representing the effects of water intrusion into buildings through openings on detention of the tsunami run-up flow. It is a distinctive point as well as the weak point of the present model because no physical evidence exists to assign a value to $C$ for each building: it is expected to vary depending on the building condition and stage of tsunami impact. However, it is also true that there must be some amount of water leakage through slits and cracks of the building side faces.

**Figure 17** presents characteristics of data scattering around the regression lines of 1:1 slope for the Case-(a) series. The mean squared error (▲) became stable for $C > 10^{-2}$ with the weak minimum at $C = 10^{-1}$, whereas the intersection value of the regression line with 1:1 slope (●, difference from the perfect agreement line) takes the minimum value of 0.8 m at $C = 10^{-2}$. Considering that the error of the maximum wave height at the coastline was 0.4–0.7 m (see **Fig. 14(a)**), the result of Case-3(a) is apparently the best among the five cases.

**Figures 18(b)** and **18(c)** respectively show the time series of water depth and flow velocity obtained using the five cases of the Case-(a) series at two points A and B in the city center, as shown in (a). Although the tsunami wave arrival time and peak values of depth and velocity depend on the value of $C$, the resultant sensitivity to $C$ does not appear to be excessive considering ambiguous factors of other kinds in numerical simulation.

Although the discussion presented above might appear to be unclear and indefinite, the overall permeable constant $C = 0.01$

for the building array before the tsunami arrival (Case-3(a)) was adopted for the following discussion because the case provided the highest correlation with measured data.

### 5.2 Tsunami effects on houses

**Figure 19** presents spatial distributions of the maximum inundation depth and the maximum flow velocity obtained from the Case-3(a) calculation, in which black rectangles show houses destroyed by tsunami waves. The water depth was greater in

the eastern part of the building collapse concentration area because the tsunami approached the city from the east, whereas higher flow velocities were found in western areas because the wavefront hitting at the end of bay intruded into open spaces and the streets directly.

Considering that the flow momentum held by a unit area water column is proportional to the multiple of depth and velocity, an indicator for tsunami run-up flow intensity, $Z$, was introduced as





$$Z = h_{\max} \times U_{\max} \, ,\tag{3}$$

where $H_{max}$ and $U_{max}$ respectively denote the maximum inundation depth and the maximum flow velocity during the flood. **Figure 20** presents a spatial distribution of $Z$, which is closely correlated with the distribution of collapsed houses colored in black in the figure.

### 5.3 Tsunami reduction effects of concrete buildings along the coast

Construction of high embankments along the coast was stated as the primary countermeasure against tsunami run-up in reconstruction guidelines issued by the Japanese Government after the 2011 Tohoku Tsunami (Cabinet Office, Government of Japan, 2011). However, such structures obstruct access to the sea, causing great inconvenience to cities with local communities mainly reliant on marine product industries.

Therefore, the effect of building arrays along the coast to control the flood flow instead of a continuous seawall was tested numerically. The white rectangles in **Fig. 21** show the trial building plot: two building layers are lined alternately to prevent seawater from flowing straight to the city center, with daily traffic given access through a hook-shaped road system. The building footprint dimensions are presented above the figure.

The color contour in **Fig. 21** shows the calculated $Z$-distribution. Compared with **Fig. 20**, the tsunami run-up flow intensity dropped drastically on streets where many wooden houses were destroyed by the 2011 Tohoku Tsunami. Results of this trial calculation suggest that properly arranged concrete buildings along the coast can serve as a seawall, reducing damage to homes behind them and sheltering some evacuation routes, although special building equipment for emergencies must be provided on lower floors, such as rigid shutters that can be closed before tsunami wave impact, and safety measures for lifeline services such as electrical power.

### 6 Conclusions

The approach presented in this paper demonstrated the possibility of accurate urban flood modeling with an internal hydraulic condition at building side faces, which allows water leakage into buildings, in the context of tsunami run-up in Kamaishi City caused by the 2011 Tohoku Earthquake. When the wall permeable constant is set to zero, the model is equivalent to a BH model. A mesh system for calculation was generated using software (ANSYS® ICEM CFD™) based on building footprints included in a digital map provide by GIAJ with a digital elevation model of 1.0 m resolution, also provided by GIAJ.

The permeable wall assumption is both the distinctive point and the weak point of the model because of the difficulty in assigning a realistic value of permeability to each building because it should be variable depending on the building condition and stage of tsunami impact. However, it is true that some amount of water leakage occurs through openings, slits, and cracks on building side faces. In this study, five values of the permeability constant $C$ defined by Eq. (3) were examined from 0 (impermeable) through $10^{-3}$, $10^{-2}$, $10^{-1}$ to 1. A comparison of computed results with field data suggests $C = 10^{-2}$ overall for cases of tsunami flooding in Kamaishi City.

Examination of time series of water depth and flow velocity at the city center showed that the consequent sensitivity on $C$ was not so great, except for a short duration around the first peak. Because accurate evaluation of hydraulic conditions at the first peak is important, further investigation is necessary to ascertain the $C$-value practically, based on results of field and experimental studies. However, considering that the original definition of the permeable constant was abstract and that the permeability model was a kind of perturbation from the building-hole model, further detailed consideration of $C$-value might be meaningless at this point.

The purpose of modelling the tsunami run-up process is not only to predict hydraulic quantities such as inundation water depth but also to propose effective measures against tsunami disasters based on calculation results. This study adopted an





indicator for run-up flow intensity: $Z = U_{max} \times H_{max}$, where $U_{max}$ and $H_{max}$ respectively stand for the maximum flow velocity and maximum water depth at each point during the flood. Results showed that the spatial mapping of $Z$-value has correlation with the distribution of houses destroyed by the tsunami flow.

Numerical tests conducted for buildings along the coast demonstrated that two lines of alternately arranged concrete buildings can prevent seawater from flowing straight into the city center, while maintaining daily traffic through a hook-shaped road system. Therefore, the present model offers great potential as a tool to support improvement of city layouts for increased safety against tsunami waves.

**Acknowledgements**

We would like to thank the academic joint research group for their extensive survey of the 2011 tsunami disaster and for sharing the valuable data obtained in their survey. We also thank the River Division of the Department of Prefectural Land Development, Iwate Prefecture for providing topographic data of the Kamaishi area for this study. Finally, we thank Prof. Takashi Nakamura of the Tokyo Institute of Technology for his assistance in compiling the program used for GPU calculation.

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

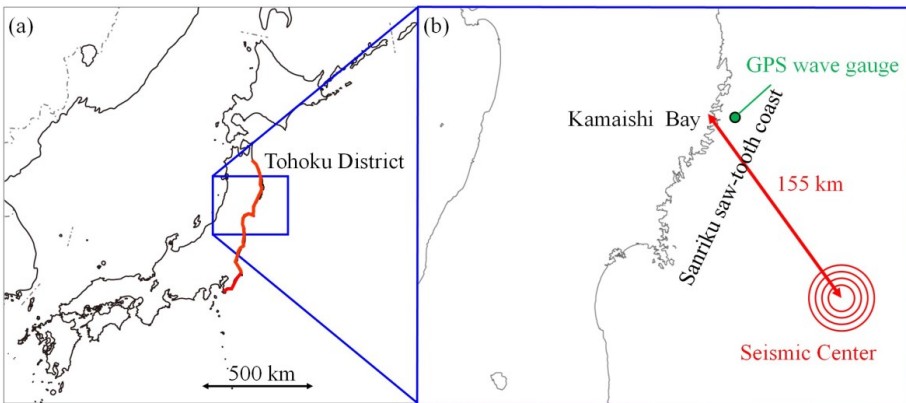

**Figure 1: Geometry of coastlines and location of study site.**
**(a) Map of Japan. Red shows the coastal region damaged by the 2011 Tohoku Tsunami.**
**(b) Locations of the seismic center, Kamaishi Bay, and the nearest GPS wave gauge.**





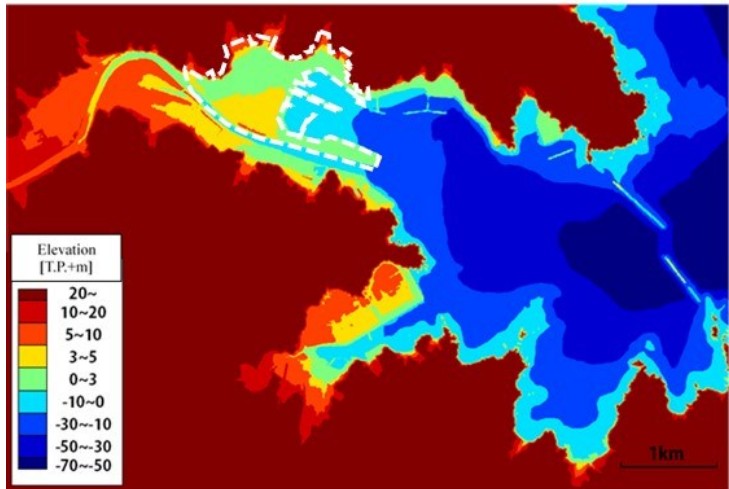

**Figure 2: Bathymetry and surrounding topography of Kamaishi Bay.**

**Tokyo Peil (T.P.)+m: Elevation above the average sea level in Tokyo Bay**

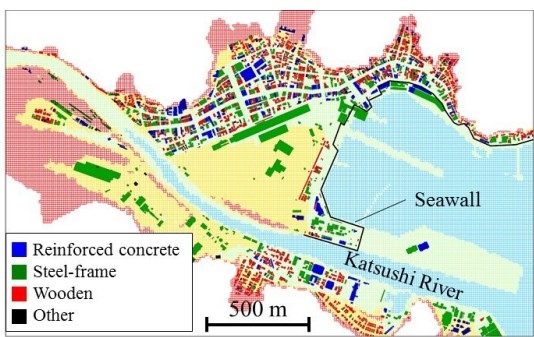

**Figure 3: Classification of building structures.**

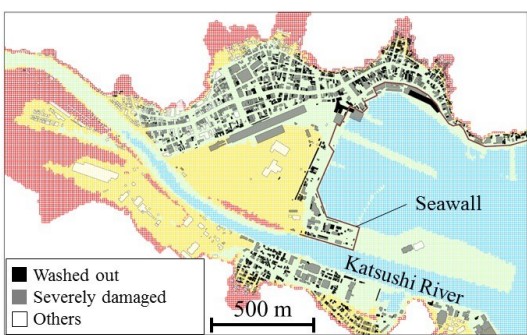

**Figure 4: Classification of building damage.**




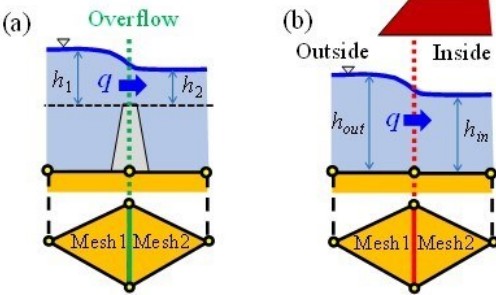

**Figure 5: Conditions of inner boundaries:**
**(a) seawalls and (b) building walls.**

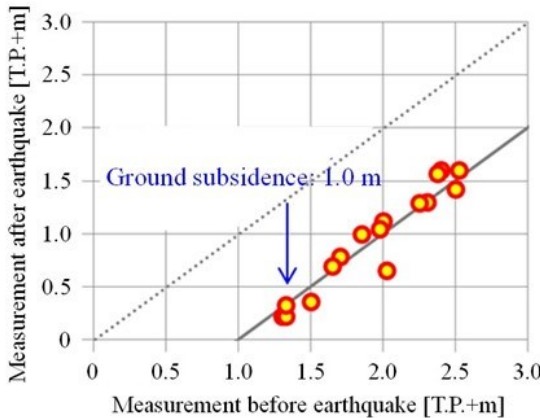

**Figure 6: Ground subsidence near the coast.**

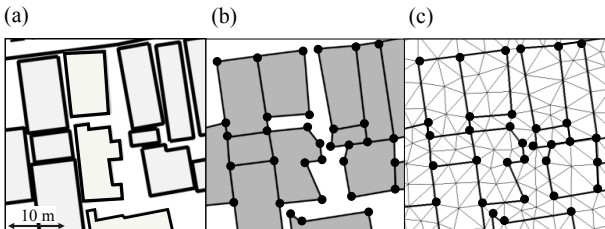

**Figure 7: Simplification and redefinition of building footprint polygons.**

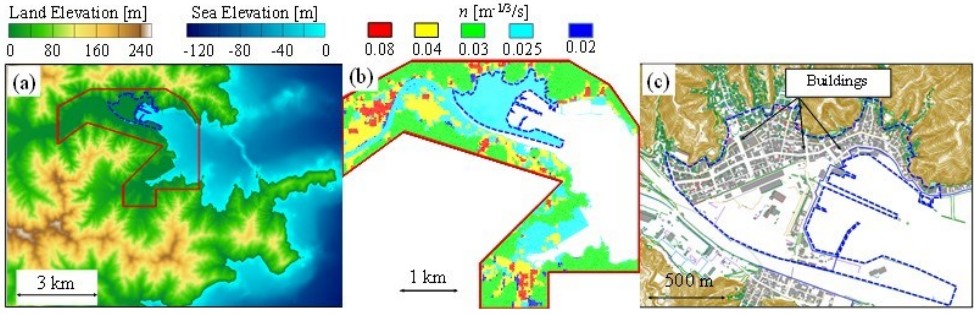





**Figure 8: Area of calculation domain:**

**(a) geometry of calculation domain in a wide view;**

**(b) Manning's roughness in calculation domain corresponding to the red line in (a); and**

**(c) domain for detailed calculation corresponding to the blue dotted line in (a).**

Table 1 Manning's roughness coefficients.

| Land use | Manning's roughness [s·m$^{-1/3}$] |
|---|---|
| Water area | 0.025 |
| Farmland | 0.02 |
| Forest | 0.03 |
| Factory site | 0.04 |
| Residential area (low density) | 0.04 |
| Residential area (high density) | 0.08 |
| Road, vacant land | 0.025 |

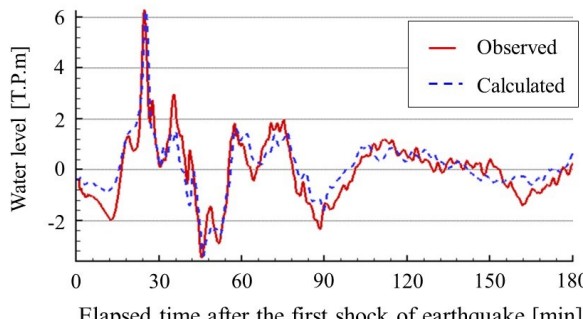

**Figure 9: Water surface displacement at the GPS wave gauge station.**

Table 2 Numerical simulation cases.

| Permeability constant, $C$ | Building layout | |
|---|---|---|
| | before tsunami | after tsunami |
| 0.0 | Case-1(a) | Case-1(b) |
| $10^{-3}$ | Case-2(a) | Case-2(b) |
| $10^{-2}$ | Case-3(a) | Case-3(b) |
| $10^{-1}$ | Case-4(a) | Case-4(b) |





| 10⁰ | Case-5(a) | Case-5(b) |

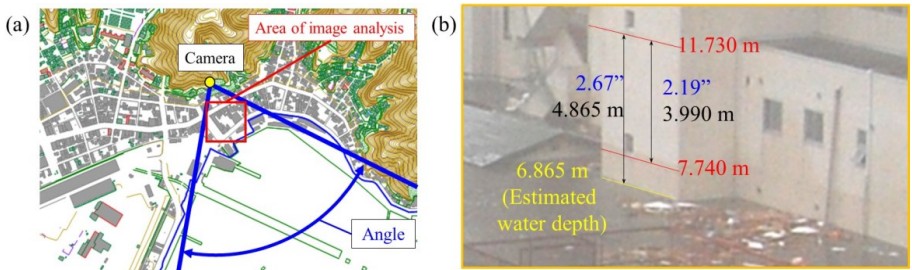

**Figure 10: Estimation of tsunami wave height near the coast:**

**(a) shooting area and (b) illustration of analysis.**

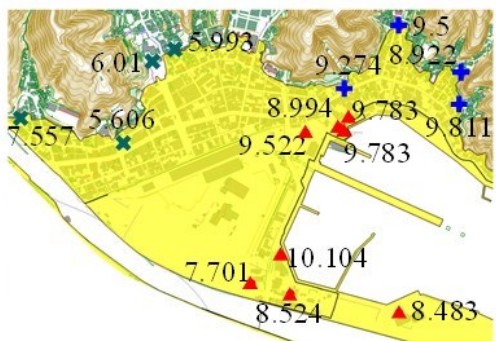

**Figure 11: Data of water surface traces (TTJS Group (2011)): plots show positions of measurements; numbers**

**show the measured height in T.P. +m.**

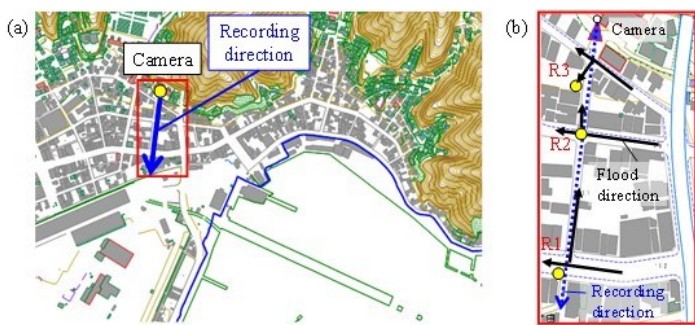

**Figure 12: Location of video recording:**

**(a) shooting direction and (b) crossings for measurement.**




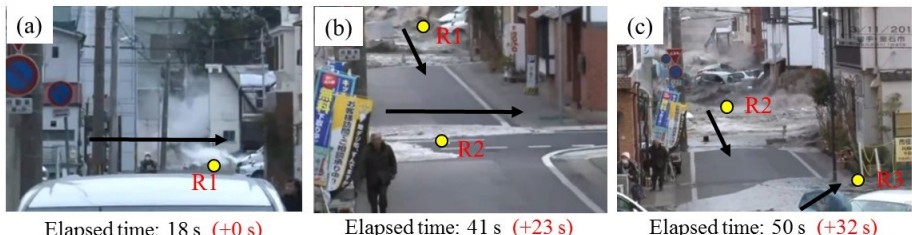

Elapsed time: 18 s  (+0 s)       Elapsed time: 41 s  (+23 s)       Elapsed time: 50 s  (+32 s)

**Figure 13: Images of wavefront passage at crossing.**

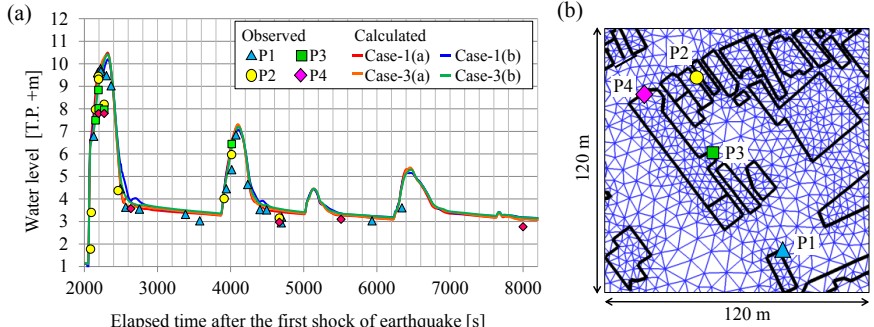

**Figure 14: Time series of water surface displacement near the coast: comparison between (a) measured and (b) calculated target points for photograph analysis.**




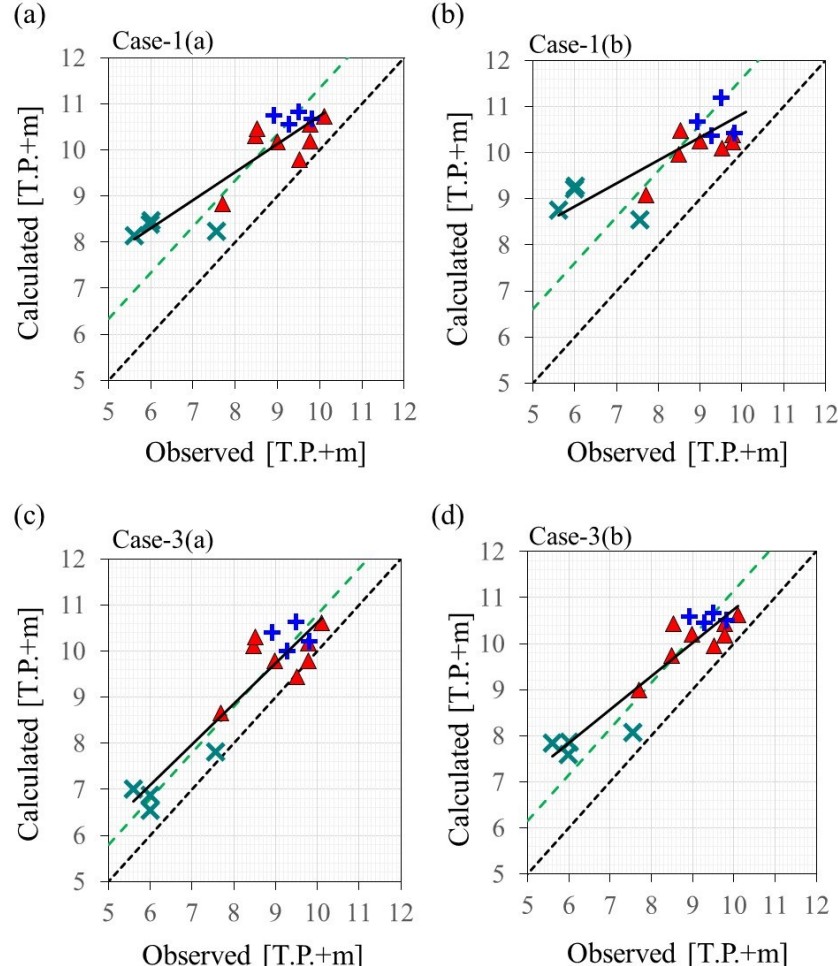

**Figure 15: Comparison between calculated and measured maximum heights. Symbols are the same as those used for Fig. 11:**

**(a) Case-1(a) – $C$=0.0, buildings before tsunami;**

**(b) Case-1(b) – $C$=0.0, buildings after tsunami;**

**(c) Case-3(a) – $C$=0.01, buildings before tsunami; and**

**(d) Case-3(b) – $C$=0.01, buildings after tsunami.**





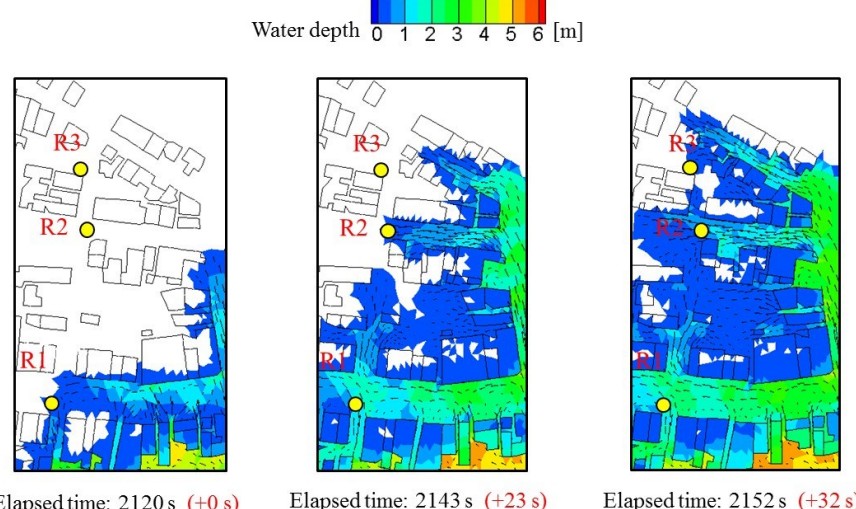

**Figure 16: Calculated wavefront propagation corresponding to the measured values in Fig. 13.**

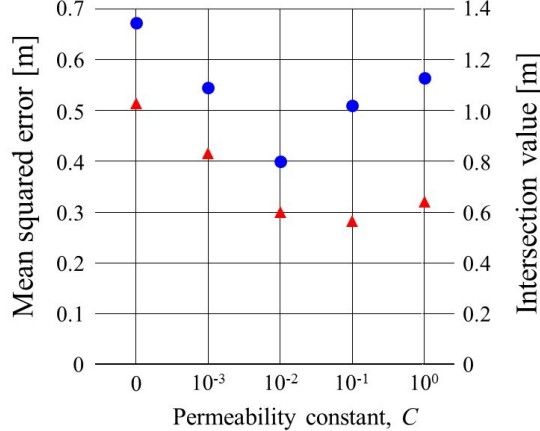

**Figure 17: Degree of regression by 1:1 slope line:**

▲, mean squared error; ●, intersection value (difference from perfect agreement).



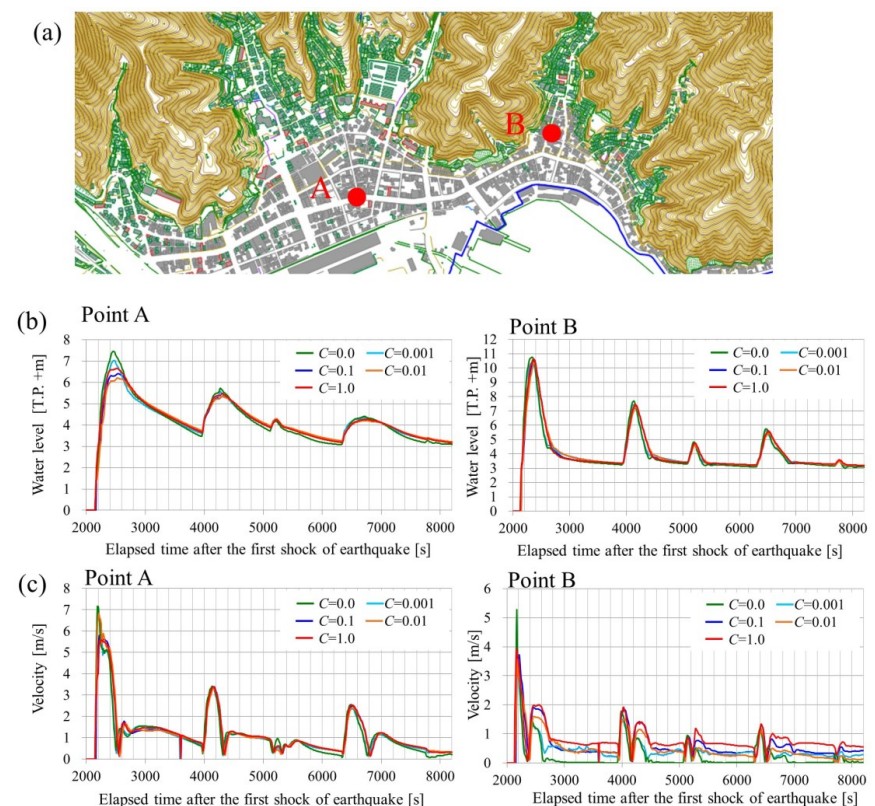

**Figure 18: Time series of flow variables at the city center:**
**(a) examination point, (b) water depth, and (c) flow velocity.**

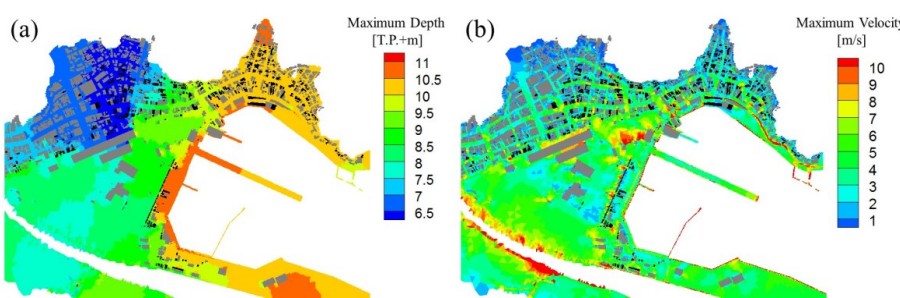

**Figure 19: Mappings of maximum depth and maximum flow velocity during flooding (Case-3(a)):**
**(a) maximum depth and (b) maximum velocity.**

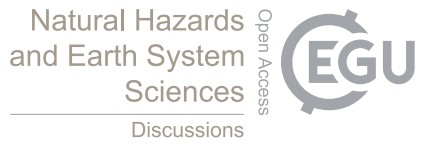



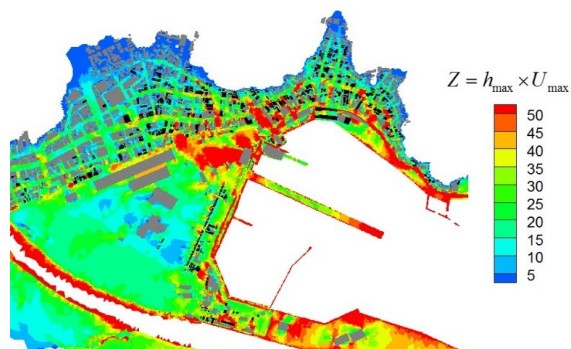

**Figure 20: Z-value mapping for the original building array (*C* = 0.01).**

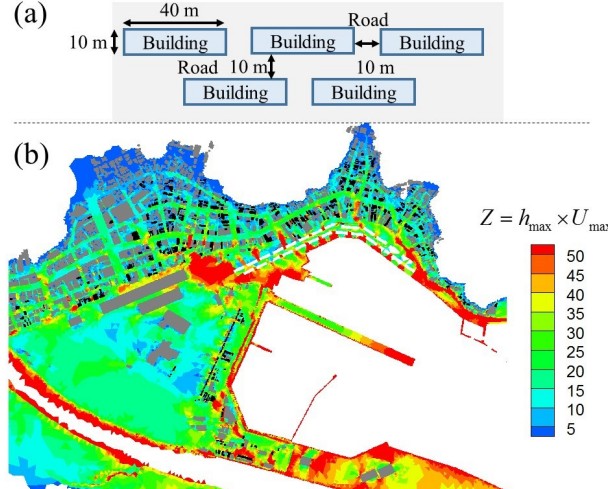

**Figure 21: Z-value mapping for the testing building plot (*C* = 0.01):**
**(a) building plot and (b) Z-value mapping.**

