# Peer review of "High-resolution modeling of tsunami run-up flooding: A case study of flooding in Kamaishi City, Japan, induced by the 2011 Tohoku Tsunami"

_Natural Hazards and Earth System Sciences, 2017_

## Referee Comment (RC1) · Anonymous Referee #1 · 24 Jul 2017

This is a very valuable work, taking flood inundation modeling to the next necessary step of highly detailed modeling where the flow can pass through buildings. This is very realistic, as buildings are indeed permeable, due to flow passing through doors, windows, and broken away walls. Furthermore, the determination of building permeability for wooden buildings will be a useful parameter for future work by practicioners and researchers. I hope this work inspires others to formalize such values for other building types as well.

In general, the English needs some work, so the paper should be corrected by a native speaker before final publication.

P2 L21-22. I do not understand the meaning of "... with 2 grid sizes...".

P2 L37. Why was Z=HU used as the indicator of flow intensity? This is flowrate. Wouldn't momentum flux HUˆ2 be a better indicator, as this is what forces on structures usually depend on? Either way, the authors should justify their choice of the parameter they choose to use.

P3 L11. Is Kamaishi really reliant on marine products? Isn't the city's main industry its factory for production of steel products?

P6 L6 you should cite the joint research group in a proper reference such as Mori N, Takahashi T, Yasuda T, Yanagisawa H. Survey of 2011 Tohoku earthquake tsunami inundation and run‐up. Geophysical research letters. 2011 Apr 1;38(7).

Table 1. The Manning's n roughness values shown look too small, especially for Forest, Factory, Residential areas. Bricker et al shows up to 0.15 for high-density urban, and greater than 0.1 for forests (up to 0.2 for dense forests with branches submerged).

P6 L13 if the local resident's video is available (i.e., YouTube), you should cite that reference here.

P6 L28 The fact that the Kamaishi bay-mouth breakwater was ignored should be justified more, as the breakwater had an effect on delaying tsunami arrival time onshore, and also mitigated flood elevation and speed onshore. See for example, Tomita et al. 2012. Effect of breakwaters on reducing flow depth during the Great East Japan Tsunami. Journal of JSCE, series B2 (Coastal Engineering).68(2):I_156-60.

Section 5.3. The protection given to inland buildings due to shielding by concrete buildings near the coast reminds me of a paper I saw by Takagi et al (2015) Assessment of the effectiveness of general breakwaters in reducing tsunami inundation in Ishinomaki. Coastal Engineering Journal. 2014 Dec;56(04):1450018. They may have discussed similar effect.

---

## Referee Comment (RC2) · Anonymous Referee #2 · 18 Aug 2017

(A formatted version of this review is attached)

The paper presents a Numerical model that is used to calculate tsunami run-up. The model is applied to study the flooding of the 2011 event in Kamaishi city, considering the buildings of the city in a high resolution numerical simulation. It is a necessary next step in the tsunami hazard assessment that is here finally addressed.

In Chapter 1 the authors explain the objective of the paper: to cover with their model the existence gap of including buildings in the simulations of tsunami run-up, by including the hydraulic effects of the presence of buildings. Following, they review the building array treatments in urban flood simulation models, revisiting the previous work on each

one of the 4 existing treatments (BR: Building resistance, BB: building blocks, BH: building holes, BP: building porosity). This study deals with the last one, assuming a permeability for the walls of the buildings.

In Chapter 2 a description of the study area is given, by means of figures including the buildings and their characteristics, specifying which of them were washed in the 2011 event.

Chapter 3 presents the numerical model development, the assumptions considered. The authors also include here the sources of the data used, the cases that were considered in the application of the numerical model and the sources of data for the model "verification".

Chapter 4 shows the results obtained in Kamaishi after the application of the developed numerical model. They show the results on 3 different parts of the flooded area: time series on several points, most elevated points reached by the flooding, and arrival time to several intersections of the city. The results are compared with the data that was recorded in several ways (cameras, flooding marks, etc.). The sources of this comparison data is explained in chapter 3 (verification).

Chapter 5 discusses the influence of the permeability constant in the numerical simulations and in the final value of the run-up and studies the effect of the presence of the buildings in the flooding processes, by means of numerical simulations of several hypothetical situations with specific layouts of the buildings near the coast. The authors introduce here an indicator for tsunami run-up flow intensity, Z=Hmax*Umax.

Finally, Chapter 6 draws some conclusions of the presented work.

GENERAL COMMENTS:

The topic is suitable for the journal since it addresses an issue which could be of interest to the scientific community. The document is up to the international standards and the length of the paper is adequate. High-resolution modeling of tsunami run-

up flooding: A case study of flooding in Kamaishi City, Japan, induced by the 2011 Tohoku Tsunami has been analysed with interesting conclusions. The results obtained with the developed numerical model present an interesting replication of the recorded data. However, some more explanations are needed in some chapters, in order make it easier the reading and understanding of the study. In addition, the introduced indicator Z, is here discussed. The reviewer would like to give some comments and suggest corrections in order to increase its overall significance.

Abstract: Although the use of U to represent the flow velocity is quite common and it is explained in the chapter 5.2, the abstract must be standalone and thus, the definition of Hmax and Umax must be given. The presence of the results of numerical simulations (lines 18-20) must be adequately presented. The addition of a sentence like "As a possible mitigation measure, the influence of the buildings in the flow has been addressed..." would increase the text flow.

1.-Introduction:

The building array treatments are widely explained. But this wide explanation distract from the objective of the paper. A briefer explanation is suggested since the references are enough to study it if necessary. In addition, and this is something common all along the paper, the structure of the chapters is not clear. The inclusion of a paragraph explaining what the reader is going to find on each chapter is needed to improve the understanding. If not, although each part is well explained the reader lose their sense of the bigger picture. In the introduction it is not mentioned that the model has been applied as well to study the influence of the concrete buildings. One of the main points of the study is the application of an alternative mitigation measure (not just a seawall) to reduce the tsunami action and to allow, at the same time, the normal work on marine industries.

3.- Methods and materials:

An introduction must be included (between 3 and 3.1) to explain what the readers are

about to find in this chapter. The characteristics of the model are well explained and referred. Is this model new or has it been presented before? If it is new it should be said clearly, or even named. In this chapter the characteristic of the numerical model, the application case data sources, and verification data sources are presented together.. These 3 different parts should be separated in order to make it easier the understanding, because they present independent parts of the study. In addition the verification data and the results can be explained together what would improve the overall understanding. This reviewer suggests the change of the structure of chapters 3 and 4 to:

______Chapter 3. The numerical model (including chapters 3.1 and 3.2).

______Chapter 4. Application case: Kamaishi port under 2011 event. 4.1 Mesh generation (including 3.3.1, 3.3.2, and 3.3.3). 4.2 Calculation condition (including 3.4)

______Chapter 5. Validation of the results. Include an introduction explaining that the results of the numerical simulations presented in the previous chapter are here presented and compared to those real data recorded. 3 comparisons: 5.1 Tsunami wave height near the coast (including 3.5.1 and 4.1). 5.2 Local highest water surface (including 3.5.2 and 4.2). 5.3 Wave front propagation on streets (including 3.5.3 and 4.3).

Again, each chapter must include an introduction.

5.- Discussion:

An introduction explaining the 2 aspects that are in this chapter (C and Z) is needed.

5.2. Here the indicator Z=U max*Hmax is presented. This is the product of the maximum inundation depth and the maximum flow velocity during the flood. However, the maximum water depth and the maximum flow velocity are not always simultaneous. The value that should be considered is Z=(U*H)max, which is the real maximum value of the product. The indicator must be recalculated or an explanation is needed to maintain the original expression. This product is used to estimate the human instability

hazard (Jonkman et al., 2008)

Jonkman, S., Vrijling, J., and Vrouwenvelder, A.: Methods for the estimation of loss of life due to floods: a literature review and a proposal for a new method, Nat. Hazards, 46, 353–389, doi:10.1007/s11069-008-9227-5, 2008.

SPECIFIC COMMENTS

Page 1 Line 10: shallow water equations

Page 1 Line 39: The reference Gallinen must be Gallien

Page 2 Line 34: permeability constant, C (from..

Page 6 Line 7: It is not included in the text the reference of the survey. In the reference chapter it is included the 2011 tohoku earthquake tsunami joint survey, but it must be referred in the text.

Page 6 Line 30: The influence of the port in the flooding was cited by Tomita in

T. Tomita, G.-S. Yeom, M. Ayugai, T. Niwa, Breakwater Effects on Tsunami Inundation Reduction in the 2011 off the Pacific Coast of Tohoku Earthquake, J. Japan Soc. Civ. Eng. Ser. B 2(Coastal Eng. 68 (2012) 4–8.

In view of this a comment on the no-consideration of the port in the simulation, as well as the citation of Tomita′s paper must be included.

Page 7 Line 10: Is this video available on the internet? If so, a reference would be interested.

Page 8 Line 1: The expression includes hmax, but in the rest of the manuscript it is called Hmax.

FIGURES:

Figure 11 is called for the first time in page 6 line10, but the symbols contained in it are not explained until Figure 15 is called in line 34. They should be explained in the foot

of the figure.

Figure 14a. In this figure are depicted the water levels at 4 points, but just the results of the model for the P3 are represented. However there are just 3 points photographed in P3. Other points have many more dots so it seems logical to depict other point time series instead of P3. In addition, the fact that all the dots (even those from other points like P1, P2 and P4) agreed fairly well in the P3 time series is important as to be highlighted.

REFERENCES:

In page 11 line17 the reference of Water and Disaster management Bureau is not included in the manuscript text

In page 5 line 23 the reference called here Central disaster prevention council, is not included in the references list.

Please also note the supplement to this comment:
https://www.nat-hazards-earth-syst-sci-discuss.net/nhess-2017-222/nhess-2017-222-RC2-supplement.pdf

---

## Author Comment (AC1) · 18 Aug 2017

[Comment-1] P2 L21-22. I do not understand the meaning of "... with 2 grid sizes...".

[Reply-1] Liu et al. (2001) showed results of two calculations with grid size of 50 m and 5.5 m, respectively, to discuss the effect of building layout resolution on tsunami run-up flow calculation for inundation caused by the 1896 Sanriku Earthquake Tsunami.

[Comment-2] P2 L37. Why was Z=HU used as the indicator of flow intensity? This is flowrate. Wouldn't momentum flux HUˆ2 be a better indicator, as this is what forces on structures usually depend on? Either way, the authors should justify their choice of the
parameter they choose to use.

[Reply-2] We adopted Z=HU as flow intensity indicator which means the momentum contained in a unit area water column in old manuscript. As the reviewer commented, however, the momentum flux (Z=HUˆ2) seems better for the indicator. Therefore, we will adapt the spatial distribution of latter in the new manuscript (Fig.20, 21). Because the new indicator showed the same tendency as the former one, the discussion in Section 5.2 was kept in the new manuscript, except the change of notation for indicator from Z to IF to avoid confusion with elevation (z).

[Comment-3] P3 L11. Is Kamaishi really reliant on marine products? Isn't the city's main industry its factory for production of steel products?

[Reply-3] The city of Kamaishi developed by the steel industry after a large iron mine was found in 1857, and had the peak of population 92,123 in 1963. In addition, the working population of the marine product industry at that time was about 2.5 times larger than that of the current. After closing the mine in 1993 and the refinery in 1998, population decreased to 35,000 at present, and its major industry became marine industry after improvement of port. We will change the sentence in the new manuscript as follows:

[Revised]: The Kamaishi City population of approximately 35,000 is mainly reliant on marine product industries and steel industry.

[Comment-4] P6 L6 you should cite the joint research group in a proper reference such as Mori N, Takahashi T, Yasuda T, Yanagisawa H. Survey of 2011 Tohoku earthquake tsunami inundation and runâËŸARËĞ up. Geophysical research letters. 2011 Apr 1;38(7).

[Reply-4] We will cite their work in the new manuscript and add the website to the reference list.

[Comment-5] Table 1. The Manning's n roughness values shown look too small, especially for Forest, Factory, Residential areas. Bricker et al shows up to 0.15 for high-density urban, and greater than 0.1 for forests (up to 0.2 for dense forests with branches submerged).

[Reply-5] Because the flow resistance by buildings is taken account as the drag force in BH model, the ground surface roughness coefficient should be smaller than BR model in which the building drag resistance is conveniently included in the surface roughness. Therefore, we adopted the smaller value for Manning's n for the "city center area where BH model was used". However, we agree to reviewer's comment that larger roughness coefficient should be taken for "surrounding areas where we adopted BR model". Therefore, we applied the values of Manning's n proposed by Bunya (2010), referring Bricker's paper for the "surrounding area" in the new manuscript. The new results did not show much difference in the "city center area" from those in the old manuscript. We will replace the new calculation results (Fig.14-19), and add Bunya's work in the text and reference list.

[Comment-6] P6 L13 if the local resident's video is available (i.e., YouTube), you should cite that reference here.

[Reply-6] We will add the URL of the website to the reference list.

[Comment-7] P6 L28 The fact that the Kamaishi bay-mouth breakwater was ignored should be justified more, as the breakwater had an effect on delaying tsunami arrival time onshore, and also mitigated flood elevation and speed onshore. See for example, Tomita et al. 2012. Effect of breakwaters on reducing flow depth during the Great East Japan Tsunami. Journal of JSCE, series B2 (Coastal Engineering).68(2):I_156-60.

[Reply-7] We agree reviewer's comment that calculation condition at the bay mouth was different from the actual situation. But, we hope the reviewer understand that the point of our paper is to consider the effect of dense building arrangement on the tsunami run-up flow. We know Tomita et al. (2012) investigated the effect of breakwater on the tsunami propagation into the bay by comparing "distinctive three calculations"; with the

breakwater before tsunami arrival; with damaged breakwater configuration measured after the tsunami; and without breakwater, while they did not show the tsunami wave deformation in the process of breakwater destruction. It is still remained for future study. Because of the uncertainness, we did the elaborate photo image analysis for tsunami wave height just near the coast line in order to examine the calculated time series near the coast line could be used for the run-up calculation in the city center area. We hope again the reviewer understand the point of this study and our efforts. We will add the purpose of the photo image analysis at the beginning of section 3.5.1 in order to make sure our consideration.

[Revised]: As mentioned earlier, in this calculation, the breakwater at the bay mouth was considered with damaged configuration measured after the tsunami due to the uncertainty of its destruction process. In this study, therefore, time series of tsunami wave height near the coast line were obtained by image analysis was carried out using digital photographs taken by residents in order to examine the calculated time series near the coast line could be used for the run-up calculation in the city center area.

[Comment-8] Section 5.3. The protection given to inland buildings due to shielding by concrete buildings near the coast reminds me of a paper I saw by Takagi et al (2015) Assessment of the effectiveness of general breakwaters in reducing tsunami inundation in Ishinomaki. Coastal Engineering Journal. 2014 Dec;56(04):1450018. They may have discussed similar effect.

[Reply-8] Takagi et al. (2014) discussed the effect of breakwater surrounding the port of Ishinomaki on the tsunami wave deformation and tsunami impact on buildings in Ishinomaki City using BR model, and their result was that breakwater did not have remarkable effect for tsunami attenuation. We will introduce their works in the text and add the paper in reference list.

PS. We will make native check before submitting final revised manuscript.

Please also note the supplement to this comment:
https://www.nat-hazards-earth-syst-sci-discuss.net/nhess-2017-222/nhess-2017-222-AC1-supplement.pdf

———————————————————
[Figure]

**Supplement:**

[Figure]

Fig. 14: Time series of water surface displacement near the coast:

comparison between (a) measured and (b) calculated

target points for photograph analysis.

[Figure]

Fig. 15: Comparison between calculated and measured maximum heights. Symbols are the same as those used for Fig. 11:

(a) Case-1(a) – $C$=0.0, buildings before tsunami;

(b) Case-1(b) – $C$=0.0, buildings after tsunami;

(c) Case-3(a) – $C$=0.01, buildings before tsunami;

(d) Case-3(b) – $C$=0.01, buildings after tsunami.

[Figure]

| Elapsed time: 2120 s (+0 s) | Elapsed time: 2143 s (+23 s) | Elapsed time: 2152 s (+32 s) |

Fig. 16: Calculated wavefront propagation corresponding to the measured values in Fig. 13.

[Figure]

Fig. 17: Degree of regression by 1:1 slope line:

▲, mean squared error; ●, intersection value (difference from perfect agreement).

[Figure]

Fig. 18: Time series of flow variables at the city center:

(a) examination point, (b) water depth, and (c) flow velocity.

[Figure]

Fig. 19: Mappings of maximum depth and maximum flow velocity during flooding (Case-3(a)):
(a) maximum depth and (b) maximum velocity.

[Figure]

Fig. 20: Z-value mapping for the original building array ($C = 0.01$).

[Figure]

Fig. 21: Z-value mapping for the testing building plot ($C = 0.01$):

(a) building plot and (b) Z-value mapping.

---

## Editor Comment (EC1) · M. Gonzalez (Editor) · 31 Aug 2017

Dear Authors,

Please, includes in your response to the second reviewer the corrected manuscript (as an attachment) including in it all your corrections and modifications.

Mauricio G. NHESS Editor

---

## Author Comment (AC2) · 4 Sep 2017

GENERAL COMMENTS: The topic is suitable for the journal since it addresses an issue which could be of interest to the scientific community. The document is up to the international standards and the length of the paper is adequate. High-resolution modeling of tsunami run-up flooding: A case study of flooding in Kamaishi City, Japan, induced by the 2011 Tohoku Tsunami has been analysed with interesting conclusions. The results obtained with the developed numerical modelãĂ Ăpresent an interesting replication of the recorded data. However, some more explanations are needed in some chapters, in order make it easier the reading and understanding of the study. In
addition, the introduced indicator Z, is here discussed. The reviewer would like to give some comments and suggest corrections in order to increase its overall significance.

Abstract: Although the use of U to represent the flow velocity is quite common and it is explained in the chapter 5.2, the abstract must be standalone and thus, the definition of Hmax and Umax must be given. The presence of the results of numerical simulations (lines 18-20) must be adequately presented. The addition of a sentence like "As a possible mitigation measure, the influence of the buildings in the flowing has been addressed..." would increase the text flow.

[Reply]

We will add the explanation of the definition of Hmax and Umax.

We will revise the sentence about the results by following your suggestion.

1.-Introduction: The building array treatments are widely explained. But this wide explanation distract from the objective of the paper. A briefer explanation is suggested since the references are enough to study it if necessary. In addition, and this is something common all along the paper, the structure of the chapters is not clear. The inclusion of a paragraph explaining what the reader is going to find on each chapter is needed to improve the understanding. If not, although each part is well explained the reader lose their sense of the bigger picture. In the introduction it is not mentioned that the model has been applied as well to study the influence of the concrete buildings. One of the main points of the study is the application of an alternative mitigation measure (not just a seawall) to reduce the tsunami action and to allow, at the same time, the normal work on marine industries.

[Reply]

In order to clarify the position of the present research, we think it is necessary to widely introduce the building array treatments. However, to avoid obscuring the purpose of this research, we will simplify the introduction of the previous study in the new manuscript.

As you pointed out, the contents discussed in 5.3 and the structure of the chapter were not described in the introduction. In the revised manuscript, we will add these description.

3.- Methods and materials: An introduction must be included (between 3 and 3.1) to explain to the reader what they are about to find in this chapter. The characteristics of the model are well explained and referred. Is this model new or has it been presented before? If it is new it should be said clearly, or even named. In this chapter the characteristic of the numerical model, the application case data sources, and verification data sources are presented together.. These 3 different parts should be separated in order to make it easier the understanding, because they present independent parts of the study. In addition the verification data and the results can be explained together what would improve the overall understanding. This reviewer suggests the change of the structure of chapters 3 and 4 to: Chapter 3. The numerical model (including chapters 3.1 and 3.2) Chapter 4. Application case: Kamaishi port under 2011 event. Introduction explaining the 2011 event 4.1 Mesh generation (including 3.3.1, 3.3.2, and 3.3.3) 4.2 Calculation condition (including 3.4) Chapter 5. Validation of the results. Include an introduction explaining that the results of the numerical simulations presented in the previous chapter are here presented and compared to those real data recorded. 3 comparisons: o 5.1 Tsunami wave height near the coast (including 3.5.1 and 4.1) 5.2 Local highest water surface (including 3.5.2 and 4.2) 5.3 Wave front propagation on streets (including 3.5.3 and 4.3)

[Reply]

Thank you for your suggestion. According to your proposal, we will change the composition of the paper. In addition, we will ask to proofread the new manuscript to a native speaker.

Again, each chapter must contain an introduction.

5.- Discussion: An introduction explaining the 2 aspects that are in this chapter (C and

[Figure]

Z) is needed.

[Reply]

We will add the introduction explaining of permeability coefficient and flow intensity indicator.

5.2. Here the indicator Z=U max*Hmax is presented. This is the product of the maximum inundation depth and the maximum flow velocity during the flood. However, the maximum water depth and the maximum flow velocity are not always simultaneous. The value that should be considered is Z=(U*H)max, which is the real maximum value of the product. The indicator must be recalculated or an explanation is needed to maintain the original expression. This product is used to estimate the human instability hazard (Jonkman et al., 2008) Jonkman, S., Vrijling, J., and Vrouwenvelder, A.: Methods for the estimation of loss of life due to floods: a literature review and a proposal for a new method, Nat. Hazards, 46, 353–389, doi:10.1007/s11069-008-9227-5, 2008.

[Reply]

We know that the maximum water depth (hmax) and maximum flow velocity are not always simultaneous. In the old manuscript, the product of the respective maximum values was used as the indicator of flow so that the value becomes large when the hmax or umax is large. Howerver, as mentioned in the reply for the comment #2 of reviewer-1, we adopted the momentum fulx (huˆ2) as the indicator in the new manuscript. In addition, it seems better to use (hmax*umaxˆ2) than (h*uˆ2)max as the maximum momentum flux. Therefore, we adopt the spatial distribution of (huˆ2)max in the new manuscript. Because the new indicator showed the same tendency as the former one, the discussion in Section 5.2 was kept in the new manuscript.

In addition, we will add the Jonkman's paper to the reference list.

SPECIFIC COMMENTS Page 1 Line 10: shallow water equations

[Reply]

We will correct the mistake in the new manuscript.

Page 1 Line 39: The reference Gallinen must be Gallien

[Reply]

We will correct the mistake in the new manuscript.

Page 2 Line 34: permeability constant, C (from..

[Reply]

We will revised in the new manuscript.

Page 6 Line 7: It is not included in the text the reference of the survey. In the reference chapter it is included the 2011 tohoku earthquake tsunami joint survey, but it must be referred in the text.

[Reply]

We will cite their work in the new manuscript and add the website to the reference list.

Page 6 Line 30: The influence of the port in the flooding was cited by Tomita in T. Tomita, G.-S. Yeom, M. Ayugai, T. Niwa, Breakwater Effects on Tsunami Inundation Reduction in the 2011 off the Pacific Coast of Tohoku Earthquake, J. Japan Soc. Civ. Eng. Ser. B 2(Coastal Eng. 68 (2012) 4–8. In view of this a comment on the no-consideration of the port in the simulation, as well as the citation of Tomita's paper must be included.

[Reply]

As mentioned in the reply of 1st reviewer's comment-7, in order to make sure our consideration for the influence of the breakwater, we will add the following sentence at the beginning of section 3.5.1 and cite Tomita's paper.

[Revised]:

As mentioned earlier, the breakwater at the bay mouth was not considered in the calculation due to the uncertainty of its destruction process. In this study, therefore, time series of tsunami wave height near the coast line were obtained by image analysis was carried out using digital photographs taken by residents in order to examine the calculated time series near the coast line could be used for the run-up calculation in the city center area.

Page 7 Line 10: Is this video available on the internet? If so, a reference would be interested.

[Reply]

We will add the URL of the website to the reference list.

Page 8 Line 1: The expression includes hmax, but in the rest of the manuscript it is called Hmax.

[Reply]

We will correct the mistake in the new manuscript.

FIGURES: Figure 11 is called for the first time in page 6 line10, but the symbols contained in it are not explained until Figure 15 is called in line 34. They should be explained in the foot of the figure.

[Reply]

We will add the explanation of the symbols in the foot of the figure.

Figure 14a. In this figure are depicted the water levels at 4 points, but just the results of the model for the P3 are represented. However there are just 3 points photographed in P3. Other points have many more dots so it seems logical to depict other point time series instead of P3. In addition, the fact that all the dots (even those from other points like P1, P2 and P4) agreed fairly well in the P3 time series is important as to be highlighted.

[Reply]

Because the center of 4 points is close to P3, we showed the calculation result at P3. In the new manuscript, we will explain the reason for using the calculation result at P3, and emphasize that the calculation results are fairly well at all 4 points.

REFERENCES: In page 11 line17 the reference of Water and Disaster management Bureau is not included in the manuscript text

[Reply]

The reference was cited to use the value of the Manning's roughness coefficient. However, in the new manuscript, we changed the value proposed by Bunya(2010), due to the suggestion from Reviewer-1. Therefore we add the following paper and delete the above reference from list.

In page 5 line 23 the reference called here Central disaster prevention council, is not included in the references list.

[Reply]

We will add the reference of Central disaster prevention council to the reference list.
* * *
[Figure]

[Figure]

Fig. 20: $I_F$-value mapping for the original building array ($C = 0.01$).

(a)

40 m  Road

10 m  Building  Building  Building

Road  10 m

10 m

Building  Building

(b)

$I_F = (hU^2)_{max}$

Fig. 21: $I_F$-value mapping for the testing building plot ($C = 0.01$):

(a) building plot and (b) $I_F$-value mapping.

**Fig. 1.**

---

## Author Comment (AC3) · 4 Sep 2017

To NHESS Editor Dear Prof. Gonzalez:

I am Ryosuke Akoh, the first author of the manuscript, "High-resolution modeling of tsunami run-up flooding: A case study of flooding in Kamaishi City, Japan, induced by the 2011 Tohoku Tsunami". Thank you for the detailed reviews of our paper. We are working on the response to reviewers' comments, and almost finished improvements of the manuscript for specific items. However, we will need some more time to respond to the second reviewer's request about changing the structure of chapters. In addition, we must need English proof check of the new manuscript by native English speaker

because of the restriction of our writing ability. We will send our total response including the answers to the second reviewer's comments in the middle of September. We hope your consideration on this matter.

Ryosuke Akoh Okayama University

---

## Author Comment (AC4) · 11 Sep 2017

Reply to Reviewer-1:

We already replied to the questions and comments made by Reviewer-1 before, but considering the comments that Reviewer-2 sent us after then, we changed the paper construction and modified the content, which affects the parts we wrote in the reply to Reviewer-1. Therefore, we sent it again.

[Comment-1]

P2 L21-22. I do not understand the meaning of "... with 2 grid sizes...".

[Figure]

[Reply-1]

Liu et al. (2001) showed results of two calculations with grid size of 50 m and 5.5 m, respectively, to discuss the effect of building layout resolution on tsunami run-up flow calculation for inundation caused by the 1896 Sanriku Earthquake Tsunami.

However, Reviewer 2 suggested that the detailed description of existing studies distracted from the objective of this paper and suggested us to reduce the introduction (Comment-1). Therefore, we eliminated the parts regarding to "two grid sizes".

[Comment-2]

P2 L37. Why was Z=HU used as the indicator of flow intensity? This is flowrate. Wouldn't momentum flux HUĖĘ2 be a better indicator, as this is what forces on structures usually depend on? Either way, the authors should justify their choice of the parameter they choose to use.

[Reply-2]

We adopted Z=HU as flow intensity indicator which means the momentum contained in a unit area water column in old manuscript. As the reviewer commented, however, the momentum flux (Z=HU2) seems better for the indicator. Therefore, we will adapt the spatial distribution of latter in the new manuscript (Fig.20, 21). Because the new indicator showed the same tendency as the former one, the discussion in Section 5.2 will be kept in the new manuscript, except the change of notation for indicator from Z to IF to avoid confusion with elevation (z).

[Comment-3]

P3 L11. Is Kamaishi really reliant on marine products? Isn't the city's main industry its factory for production of steel products?

[Reply-3]

The city of Kamaishi developed by the steel industry after a large iron mine was found

in 1857, and had the peak of population 92,123 in 1963. In addition, the working population of the marine product industry at that time was about 2.5 times larger than that of the current. After closing the mine in 1993 and the refinery in 1998, population decreased to 35,000 at present, and its major industry became marine industry after improvement of port. We will change the sentence in the new manuscript as follows:

[Revise]:

The Kamaishi City population of approximately 35,000 is mainly reliant on marine product industries and steel industry.

[Comment-4]

P6 L6 you should cite the joint research group in a proper reference such as Mori N, Takahashi T, Yasuda T, Yanagisawa H. Survey of 2011 Tohoku earthquake tsunami inundation and runâĚŸARĚĞ up. Geophysical research letters. 2011 Apr 1;38(7).

[Reply-4]

We will cite their work in the new manuscript and add the website to the reference list.

[Comment-5]

Table 1. The Manning's n roughness values shown look too small, especially for Forest, Factory, Residential areas. Bricker et al shows up to 0.15 for high-density urban, and greater than 0.1 for forests (up to 0.2 for dense forests with branches submerged).

[Reply-5]

Because the flow resistance by buildings is taken account as the drag force in BH model, the ground surface roughness coefficient should be smaller than BR model in which the building drag resistance is conveniently included in the surface roughness. Therefore, we adopted the smaller value for Manning's n for the "city center area where BH model was used". However, we agree to reviewer's comment that larger roughness coefficient should be taken for "surrounding areas where we adopted BR model".

Therefore, we applied the values of Manning's n proposed by Bunya (2010), referring Bricker's paper for the "surrounding area" in the new manuscript. The new results did not show much difference in the "city center area" from those in the old manuscript. We will replace the new calculation results (Fig.14-19), and add Bunya's work in the text and reference list.

[Comment-6]

P6 L13 if the local resident's video is available (i.e., YouTube), you should cite that reference here.

[Reply-6]

We will add the URL of the website to the reference list.

[Comment-7]

P6 L28 The fact that the Kamaishi bay-mouth breakwater was ignored should be justified more, as the breakwater had an effect on delaying tsunami arrival time onshore, and also mitigated flood elevation and speed onshore. See for example, Tomita et al. 2012. Effect of breakwaters on reducing flow depth during the Great East Japan Tsunami. Journal of JSCE, series B2 (Coastal Engineering).68(2):I_156-60.

[Reply-7]

We agree reviewer's comment that calculation condition at the bay mouth was different from the actual situation. But, we hope the reviewer understand that the point of our paper is to consider the effect of dense building arrangement on the tsunami run-up flow. We know Tomita et al. (2012) investigated the effect of breakwater on the tsunami propagation into the bay by comparing "distinctive three calculations"; with the breakwater before tsunami arrival; with damaged breakwater configuration measured after the tsunami; and without breakwater, while they did not show the tsunami wave deformation in the process of breakwater destruction. It is still remained for future study. Because of the uncertainness, we did the elaborate photo image analysis for tsunami

wave height just near the coast line in order to examine the calculated time series near the coast line could be used for the run-up calculation in the city center area. We hope again the reviewer understand the point of this study and our efforts. We will add the following sentence at the end of "3.2.1 Ground surface elevation" in the new manuscript in order to make clear that the tsunami propagation during the collapse of breakwater is still remained for future study.

[Revised]

Tomita et al. (2012) investigated the effect of breakwater on the tsunami propagation into the bay by comparing three calculations; with the breakwater before tsunami arrival; with damaged breakwater configuration measured after the tsunami; and without breakwater, while the actual process of breakwater destruction is still remained for future study. Therefore in this study, the damaged configuration measured after the tsunami (****, 20**) was assumed for calculation.

We will add the purpose of the photo image analysis at the beginning of section "4.1.1 Field data analysis" in the new manuscript in order to make sure our consideration.

[Revised]

As mentioned earlier, in this calculation, the breakwater at the bay mouth was considered with damaged configuration measured after the tsunami due to the uncertainty of its destruction process. In this study, therefore, time series of tsunami wave height near the coast line were obtained by image analysis was carried out using digital photographs taken by residents in order to examine the calculated time series near the coast line could be used for the run-up calculation in the city center area.

[Comment-8]

Section 5.3. The protection given to inland buildings due to shielding by concrete buildings near the coast reminds me of a paper I saw by Takagi et al (2015) Assessment of the effectiveness of general breakwaters in reducing tsunami inundation in Ishinomaki.

[Figure]

Coastal Engineering Journal. 2014 Dec;56(04):1450018. They may have discussed similar effect.

[Reply-8]

We guess the year of publication by Takagi et al. was "2014" though the reviewer-1 wrote "2015". In our understanding, the main topic of their numerical study using BR model was the tsunami attenuation by breakwater surrounding the port of Ishinomaki. In the same paper, they suggested that the damage of houses was smaller behind a large concrete building "from aerial photograph observation", but it was "not from numerical simulation"; their calculation was based on BR model, which could not estimate the effect of each building footprint. We will insert the following sentence in 5.3.

Some reports suggested that large buildings protected the houses behind from tsunami impact (e. g., Matsutomi et al, 2012; Takagi et al., 2014)

PS. We will make native check before submitting final revised manuscript.

---

## Author Comment (AC5) · 11 Sep 2017

GENERAL COMMENTS: The topic is suitable for the journal since it addresses an issue which could be of interest to the scientific community. The document is up to the international standards and the length of the paper is adequate. High-resolution modeling of tsunami run-up flooding: A case study of flooding in Kamaishi City, Japan, induced by the 2011 Tohoku Tsunami has been analysed with interesting conclusions. The results obtained with the developed numerical modelãĂĂpresent an interesting replication of the recorded data. However, some more explanations are needed in some chapters, in order make it easier the reading and understanding of the study. In

addition, the introduced indicator Z, is here discussed. The reviewer would like to give some comments and suggest corrections in order to increase its overall significance.

[Comment-1]

Abstract: Although the use of U to represent the flow velocity is quite common and it is explained in the chapter 5.2, the abstract must be standalone and thus, the definition of Hmax and Umax must be given.

[Reply]

We will add the definitions of the variables. Following another reviewer (#1), we changed the flow intensity indicator to (hU2)Max, the maximum of momentum flux, and we will add the explanation of h and U in the abstract, too.

The presence of the results of numerical simulations (lines 18-20) must be adequately presented. The addition of a sentence like "As a possible mitigation measure, the influence of the buildings in the flowing has been addressed..." would increase the text flow.

[Reply]

Following your suggestion, we will revise the sentence about the results. The English will be checked by an English native speaker before submitting the final manuscript.

As a possible mitigation measure, the influence of the buildings in the flowing has been addressed by a numerical experiment for solid buildings arrayed alternately in two lines along the coast. The results show that the buildings can prevent seawater from flowing straight to the city center while maintaining access to the sea.

[Revised] P.1 Abstract

[Comment-2]

1.-Introduction: The building array treatments are widely explained. But this wide explanation distract from the objective of the paper. A briefer explanation is suggested since the references are enough to study it if necessary. In addition, and this is something common all along the paper, the structure of the chapters is not clear. The inclusion of a paragraph explaining what the reader is going to find on each chapter is needed to improve the understanding. If not, although each part is well explained the reader lose their sense of the bigger picture.

[Reply]

We compacted the description of building array treatments and introduction of existing studies, and added the introduction of chapter-structure at the end. The new introduction will be checked by an English native speaker before submitting final manuscript.

In the introduction it is not mentioned that the model has been applied as well to study the influence of the concrete buildings. One of the main points of the study is the application of an alternative mitigation measure (not just a seawall) to reduce the tsunami action and to allow, at the same time, the normal work on marine industries.

[Reply]

We added a sentence about the numerical experiment on the influence of buildings along the coast on tsunami intrusion into the city.

Recent urbanization of low-lying coastal areas has increased the potential for property damage, human injury, and death caused by tsunamis. Visual data obtained during the tsunami run-up revealed that arrays of structures in urban areas induced large wave deformation and swift currents on streets, and that the currents washed objects such as garbage, cars, and debris from damaged structures, causing even more damage than tsunami run-up over uniform ground. Prediction of swift currents in urban areas by numerical flow simulation is expected to be important for evacuation programs and for city layout planning measures to mitigate tsunami damage.

Tsunami simulation models for forecasting wave propagation and deformation from the

seismic center to the coast have been developed and improved for decades. These models for high-speed calculations in a wide water body are often based on a set of shallow-water equations on a structured rectangular grid system (Imamura 1995). Models with a rectangular grid system were extended to calculate the tsunami run-up on land by formulating the wavefront propagation on a dry bed (TiTov et al. 1995, 1998; Synolakis et al. 2008). However, the tsunami run-up simulation described above requires more precise flow modeling by introducing the hydraulic effects of building arrangement.

Building array treatments in urban flood inundation models are classifiable into four types (Schubert et al. 2008; Schubert et al. 2012): building-resistance models (BR), in which large surface roughness is assigned to cells that fall within a building footprint (Liang et al. 2007) or developed parcels (Gallegos et al. 2009; Gallien et al. 2011); building-block models (BB), in which spatially distributed ground elevation data are raised to roof-top height (Brown et al. 2007; Hunter et al. 2008; Schubert et al. 2008); building–hole models (BH), in which building footprints are excluded from the flow calculation area with a free-slip wall boundary condition (Aronica et al. 1998; Aronica et al. 2005; Schubert et al. 2008); and building-porosity models (BP), in which the impact of buildings in a street block is expressed approximately by porosity and a drag coefficient in a street block (Guinot 2012; Sanders et al. 2008; Soares-Frazão et al. 2008).

The BR model is commonly adopted for tsunami run-up simulations (Gayer et al. 2010; Kaiser 2011; Suppasri et al. 2011; Bricker et al. 2015), although the model developers did not predict the velocity field. Komatsu et al.(2010), Conde et al. (2013) and Imai et al. (2013) applied the BB model for the tsunami flooding in Kota Banda Aceh of Indonesia caused by 2004 off the Indian Coast of Sumatra Island Earthquake, in two cities of Portugal caused by the 1755 Lisbon Tsunami, and in Kochi city of Japan by a historical tsunami run-up in 1707, respectively. Liu et al. (2001) applied the BH model to tsunami run-up flow caused by the 1896 Sanriku Earthquake Tsunami. Akoh

et al. (2014) proposed a permeable wall model equivalent to the BH model when the permeability constant was zero, and applied the model to the tsunami flooding in Kamaishi city of Japan during the 2011 off the Pacific Coast of Tohoku Earthquake (2011 Tohoku Tsunami hereinafter). For BP model, no report of the relevant literature has described for tsunami run-up simulation, probably because it is not easy to identify the values of porosity and building drag coefficient for respective street blocks.

For this study, the permeable wall model based on shallow flow equations proposed by Akoh et al. (2014) was used to investigate tsunami run-up details in the Kamaishi city induced by 2011 Tohoku Tsunami using more field data than used in the earlier study. Chapter-2 describes the numerical simulation method; basic formulations and building array treatment. Chapter-3 was devoted to explain numerical modeling the tsunami flooding in Kamaishi City; explanation of study site, data sources for modeling, mesh generation, and calculation conditions. Calculation results are displayed in Chapter-4 with validation data. In Chapter-5, after discussing the influence of permeability constant on calculation results, the tsunami impacts on houses is examined by introducing an indicator, IF = (hU2)max, where h and U respectively denote the water depth and the flow velocity at each point, and the effect of rigid building arrays along the coast on the reduction of IF in the city center was tested numerically, as a possible mitigation measure instead of high continuous embankments which prevents the access to the sea.

[Comment-3]

3.- Methods and materials:

An introduction must be included (between 3 and 3.1) to explain to the reader what they are about to find in this chapter.

[Reply]

Following [Comment 4] of reviewer-2, we moved Site Description (Chapter-2 in the old

manuscript) to the beginning of Application (Chapter-3 in the new manuscript). The Method and materials (Chapter-3 in the old manuscript) will become Chapter-2 in the new manuscript. In addition, we will change the chapter title "Methods" because we move 3.2 – 3.4 (old manuscript) to the chapter for validation of calculation results your next suggestion. We will insert a short sentence between 2 and 2.1 as follows:

Considering the openings of wooden houses such as doors, windows or cracks and slits cause by tsunami impacts, shallow water BH model was improved to express the effects of wall permeability by introducing the "assumption of internal hydraulic conditions" on the line segments where the walls were located. The seawall overtopping was considered in a same way.

The characteristics of the model are well explained and referred. Is this model new or has it been presented before? If it is new it should be said clearly, or even named.

[Reply]

This is a new model. We will emphasize our original idea "assumption with internal hydraulic conditions" with double quotation marks in the introduction for Chapter-2 (see the above sentence).

In this chapter the characteristic of the numerical model, the application case data sources, and verification data sources are presented together.. These 3 different parts should be separated in order to make it easier the understanding, because they present independent parts of the study. In addition the verification data and the results can be explained together what would improve the overall understanding. This reviewer suggests the change of the structure of chapters 3 and 4 to:

Chapter 3. The numerical model (including chapters 3.1 and 3.2)

Chapter 4. Application case: Kamaishi port under 2011 event.

Introduction explaining the 2011 event

4.1 Mesh generation (including 3.3.1, 3.3.2, and 3.3.3)

4.2 Calculation condition (including 3.4)

Chapter 5. Validation of the results. Include an introduction explaining that the results of the numerical simulations presented in the previous chapter are here presented and compared to those real data recorded. 3 comparisons:

5.1 Tsunami wave height near the coast (including 3.5.1 and 4.1)

5.2 Local highest water surface (including 3.5.2 and 4.2)

5.3 Wave front propagation on streets (including 3.5.3 and 4.3)

[Reply]

Thank you for your comment. We thought in the first draft as you suggested. We will change the structure of final manuscript as follows:

1. Introduction

2. Method: 2.1 Numerical model, 2.2 Assumption of internal boundary conditions.

3. Application case: 3.1 Site description, 3.2 Model set-up (3.2.1 Topographical conditions, 3.2.2 Seawalls and building footprints, 3.2.3 Mesh generation), 3.3. Hydraulic condition for calculation.

4. Validation of results: 4.1 Tsunami wave height near the coast (4.1.1 Field data analysis, 4.1.2 Verification of results), 4.2. Local highest water surface (4.2.1 Field data sources, 4.2.2 Verification of results), 4.3 Wave front propagation on streets (4.3.1 Field data analysis, 4.3.2 Verification of results).

5. Discussion: 5.1. Permeability constant, 5.2. Tsunami effects on houses, 5.3. Tsunami reduction effects concrete buildings along the coast.

6. Conclusions

Again, each chapter must contain an introduction.

[Reply]

We will insert a short introduction for each chapter. These parts will be checked by an English native speaker before submitting the final manuscript.

[Chapter 3: Application case]

The 2011 off the Pacific Coast of Tohoku Earthquake with 9.0 in seismic magnitude hit the Northeast Pacific Coast of Japan on March 11, 2011. The total death toll including still missing reached about 18,000, 90 % of whom were killed by tsunami after the earthquake in low-lying urban areas on the coast. Kamaishi City was one of the most severely damaged municipalities. (It will be followed by "3.1 Site description".)

[Chapter 4: Validation of results]

Many kinds of data were collected by the academic groups, the government and municipalities after the earthquake. The results obtained from the numerical simulation described in the previous chapter are here presented and compared to those real data.

[Chapter 5: Discussion]

The reasonable value of C, unknown parameter in the model, is discussed in this chapter based on observed data presented in the previous chapter. Then, the tsunami impacts on houses is estimated by introducing an indicator for tsunami run-up intensity, and effect of rigid building arrays along the coast is tested numerically, as a possible mitigation measure to reduce the hydraulic impact indicator in the city center.

5.- Discussion:

An introduction explaining the 2 aspects that are in this chapter (C and Z) is needed.

[Reply]

We will add the introduction as written above.

5.2. Here the indicator Z=U max*Hmax is presented. This is the product of the maximum inundation depth and the maximum flow velocity during the flood. However, the maximum water depth and the maximum flow velocity are not always simultaneous. The value that should be considered is Z=(U*H)max, which is the real maximum value of the product. The indicator must be recalculated or an explanation is needed to maintain the original expression.

This product is used to estimate the human instability hazard (Jonkman et al., 2008) Jonkman, S., Vrijling, J., and Vrouwenvelder, A.: Methods for the estimation of loss of life due to floods: a literature review and a proposal for a new method, Nat. Hazards, 46, 353–389, doi:10.1007/s11069-008-9227-5, 2008.

[Reply]

Thank you for the suggestion. As mentioned in the reply for the comment #2 of reviewer-1, we finally adopted (hU2)Max, which is flow momentum flux, for flow intensity indicator. The spatial distribution characteristics of the new indicator was basically same as the old indicator, and the points in discussion will be same as before.

SPECIFIC COMMENTS

Page 1 Line 10: shallow water equations

[Reply]

We will correct the mistake in the new manuscript.

Page 1 Line 39: The reference Gallinen must be Gallien

[Reply]

We will correct the mistake in the new manuscript.

Page 2 Line 34: permeability constant, C (from..

[Reply]

This part was eliminated from the introduction.

Page 6 Line 7: It is not included in the text the reference of the survey. In the reference chapter it is included the 2011 tohoku earthquake tsunami joint survey, but it must be referred in the text.

[Reply]

We will cite their work in the new manuscript and add the website to the reference list.

Page 6 Line 30: The influence of the port in the flooding was cited by Tomita in T. Tomita, G.-S. Yeom, M. Ayugai, T. Niwa, Breakwater Effects on Tsunami Inundation Reduction in the 2011 off the Pacific Coast of Tohoku Earthquake, J. Japan Soc. Civ. Eng. Ser. B 2(Coastal Eng. 68 (2012) 4–8. In view of this a comment on the no-consideration of the port in the simulation, as well as the citation of Tomita's paper must be included.

[Reply]

We will add the following sentence at the beginning of section "3.2.1 Topographical conditions", and will cite Tomita's paper in reference list.

[Revised]:

Tomita et al. (2012) investigated the effect of breakwater on the tsunami propagation into the bay by comparing three calculations; with the breakwater before tsunami arrival; with damaged breakwater configuration measured after the tsunami; and without breakwater, while the actual process of breakwater destruction is still remained for future study. Therefore in this study, the damaged configuration measured after the tsunami (Takahashi et al. 2011) was assumed for calculation.

We also add the following sentence at the beginning of section "4.1.1 Field data analysis" to express our consideration about the breakwater destruction.

As mentioned earlier, the breakwater at the bay mouth was considered with damaged

configuration measured after the tsunami due to the uncertainty of its destruction process. In this study, therefore, time series of tsunami wave height near the coast line were obtained by image analysis of digital photographs taken by residents in order to examine the calculated time series near the coast line could be used for the run-up calculation in the city center area.

Page 7 Line 10: Is this video available on the internet? If so, a reference would be interested.

[Reply]

We will add the URL of the website to the reference list.

Page 8 Line 1: The expression includes hmax, but in the rest of the manuscript it is called Hmax.

[Reply]

Equation (3) will be changed to (hU2)Max, as mentioned before, and we will write correctly.

FIGURES: Figure 11 is called for the first time in page 6 line10, but the symbols contained in it are not explained until Figure 15 is called in line 34. They should be explained in the foot of the figure.

[Reply]

We will add the explanation of the symbols in the foot of the figure.

Figure 14a. In this figure are depicted the water levels at 4 points, but just the results of the model for the P3 are represented. However there are just 3 points photographed in P3. Other points have many more dots so it seems logical to depict other point time series instead of P3. In addition, the fact that all the dots (even those from other points like P1, P2 and P4) agreed fairly well in the P3 time series is important as to be highlighted.

[Reply]

Calculated water levels at the four points were almost same because they were very close to one another. Therefore, we showed the calculation result at P3 which was at the center. We will mention it in the new manuscript.

The P3 located at the center of measurement area was selected for the plotting of calculation results because the four points were very close to one another and calculation results were almost the same.

REFERENCES: In page 11 line17 the reference of Water and Disaster management Bureau is not included in the manuscript text

[Reply]

The reference was cited for the Manning's roughness assumption listed in Table 1in the old manuscript. However, in the new manuscript, we adopted the values proposed by Bunya (2010) and recalculated, following the suggestion from Reviewer-1. Therefore we will add the Bunya's paper in the reference list.

In page 5 line 23 the reference called here Central disaster prevention council, is not included in the references list.

[Reply]

We will add the reference of Central disaster prevention council to the reference list.

---

## Author Comment (AC6) · 11 Sep 2017

To NHESS Editor Dear Prof. Gonzalez:

I am Ryosuke Akoh, the first author of the manuscript, "High-resolution modeling of tsunami run-up flooding: A case study of flooding in Kamaishi City, Japan, induced by the 2011 Tohoku Tsunami". Thank you for the detailed reviews of our paper. Please find our reply to the two reviewers. We already send reply to the reviewer-1 in August, but because of the manuscript structure change suggested by the reviewer-2, some parts we wrote in the reply to reviewer-1 was affected. Therefore, we send the replies to the both reviewers this time. We are now asking an English native speaker to

check our writing, and it will be completed at the middle of September. We hope your consideration on this matter.

Ryosuke Akoh Okayama University
* * *

---

## Author Response (AR1)

[revised manuscript text omitted]

**(a) geometry of calculation domain in a wide view,**

**(b) Manning's roughness in calculation domain corresponding to the red line in (a), and**

**(c) domain for detailed calculation corresponding to the blue dotted line in (a).**

**Table 1 Manning's roughness coefficients.**

| Land use | Manning's roughness [s·m$^{-1/3}$] |
|---|---|
| Water area | 0.025 |
| Farmland | 0.04 |
| Forest | 0.16 |
| Factory site | 0.05 |
| Residential area (low density) | 0.05 |
| Residential area (high density) | 0.15 |
| Road, vacant land | 0.025 |

[Figure]

**Figure 9: Water surface displacement at the GPS wave gauge station.**

**Table 2 Numerical simulation cases.**

| Permeability constant, $C$ | Building layout | |
|---|---|---|
| | before tsunami | after tsunami |
| 0.0 | Case-1(a) | Case-1(b) |
| $10^{-3}$ | Case-2(a) | Case-2(b) |
| $10^{-2}$ | Case-3(a) | Case-3(b) |
| $10^{-1}$ | Case-4(a) | Case-4(b) |
| $10^{0}$ | Case-5(a) | Case-5(b) |

[Figure]

**Figure 10: Estimation of tsunami wave height near the coast:**
**(a) shooting area and (b) illustration of analysis.**

[Figure]

**Figure 11: Time series of water surface displacement near the coast:**
**comparison between (a) measured and (b) calculated target points for photograph analysis.**

[Figure]

**Figure 12: Data of water surface traces (TTJS Group (2011)): plots show positions of measurements; numbers show the measured height in T.P. +m; ▲, +, and × show measurement locations; the numbers are maximum water levels.**

[Figure]

**Figure 13: Comparison between calculated and measured maximum heights. Symbols are the same as those used for Fig. 11:**

**(a) Case-1(a) – *C*=0.0, buildings before tsunami;**

**(b) Case-1(b) – *C*=0.0, buildings after tsunami;**

**(c) Case-3(a) – *C*=0.01, buildings before tsunami; and**

**(d) Case-3(b) – *C*=0.01, buildings after tsunami.**

[Figure]

**Figure 14: Location of video recording:**

**(a) shooting direction and (b) crossings for measurement.**

[Figure]

**Figure 15: Images of wavefront passage at crossing.**

[Figure]

**Figure 16: Calculated wavefront propagation corresponding to the measured values in Fig. 13.**

[Figure]

**Figure 17: Degree of regression by 1:1 slope line:**

▲, mean squared error; ●, intersection value (difference from perfect agreement).

[Figure]

**Figure 18: Time series of flow variables at the city center:**
**(a) examination point, (b) water depth, and (c) flow velocity.**

[Figure]

**Figure 19: Mappings of maximum depth and maximum flow velocity during flooding (Case-3(a)): (a) maximum depth and (b) maximum velocity.**

[Figure]

$$I_F = (hU^2)_{max}$$

**Figure 20: $I_F$ -value mapping for the original building array ($C = 0.01$).**

[Figure]

**Figure 21:** $I_F$ **-value mapping for the testing building plot ($C = 0.01$):**
**(a) building plot and (b)** $I_F$ **-value mapping.**

**Response to Reviewer-1:**

[Reviewer's Comment: Black, Author's Comment: Blue, Author's changes in manuscript : Red]

[Comment-1]

P2 L21-22. I do not understand the meaning of "... with 2 grid sizes...".

[Reply-1]

Liu et al. (2001) showed results of two calculations with grid size of 50 m and 5.5 m, respectively, to discuss the effect of building layout resolution on tsunami run-up flow calculation for inundation caused by the 1896 Sanriku Earthquake Tsunami. However, Reviewer 2 suggested that the detailed description of existing studies distracted from the objective of this paper and suggested us to reduce the introduction (Comment-1). Therefore, we eliminated the parts regarding to "two grid sizes".

[Revised]

**Page 2 Line 12-13**

Liu et al. (2001) applied the BH model to tsunami run-up flow caused by the 1896 Sanriku Earthquake Tsunami.

[Comment-2]

P2 L37. Why was Z=HU used as the indicator of flow intensity? This is flowrate. Wouldn't momentum flux HUˆ2 be a better indicator, as this is what forces on structures usually depend on? Either way, the authors should justify their choice of the parameter they choose to use.

[Reply-2]

We adopted $Z=HU$ as flow intensity indicator which means the momentum contained in a unit area water column in old manuscript. As the reviewer commented, however, the momentum flux ($Z=HU^2$) seems better for the indicator. Therefore, we will adapt the spatial distribution of latter in the new manuscript (Fig.20, 21). Because the new indicator showed the same tendency as the former one, the discussion in Section 5.2 is kept in the new manuscript, except the change of notation for indicator from $Z$ to $I_F$ to avoid confusion with elevation ($z$).

[Revised]

**Page 1 Line 18**

$I_F = (hU^2)_{max}$,

**Page 2 Line 25,Line 26**

$I_F = (hU^2)_{max}$

**Page 8 Line 12**

$I_F$,

**Page 8 Line 13**

Equation (3)

**Page 8 Line 15**

$I_F$,

**Page 8 Line 27,**

$I_F$-distribution

**Page 9 Line 17**

$I_F = (hU^2)_{max}$

**Page 9 Line 18**

$I_F$-value

[Comment-3]

P3 L11. Is Kamaishi really reliant on marine products? Isn't the city's main industry its factory for production of steel products?

10   [Reply-3]

The city of Kamaishi developed by the steel industry after a large iron mine was found in 1857, and had the peak of population 92,123 in 1963. In addition, the working population of the marine product industry at that time was about 2.5 times larger than that of the current. After closing the mine in 1993 and the refinery in 1998, population decreased to 35,000 at present, and its major industry became marine industry after improvement of port. We changed the sentence in the new manuscript as follows:

15   [Revised]

**Page 4 Line 7-8**

 The Kamaishi City population of approximately 35,000 is mainly reliant on marine product industries and steel industry.

[Comment-4]

20   P6 L6 you should cite the joint research group in a proper reference such as Mori N, Takahashi T, Yasuda T, Yanagisawa H. Survey of 2011 Tohoku earthquake tsunami inundation and runâ˘ARˇ up. Geophysical research letters. 2011 Apr 1;38(7).

[Reply-4]

We cited their work in the new manuscript and added the website to the reference list.

25   [Revised]

**Page 6 Line 31-32**

 An academic joint research group was organized to conduct an extensive survey of the disaster caused by the 2011 Tohoku Tsunami (TTJS Group 2011; Mori et al. 2011).

**Page 11 Line 22-23**

30   Mori N, Takahashi T, Yasuda T, and Yanagisawa H. Survey of 2011 Tohoku earthquake tsunami inundation and run-up. Geophysical Research Letters. 2011 Apr 1;38(7).

[Comment-5]

Table 1. The Manning's n roughness values shown look too small, especially for Forest, Factory, Residential areas. Bricker et

35   al shows up to 0.15 for high-density urban, and greater than 0.1 for forests (up to 0.2 for dense forests with branches submerged).

[Reply-5]

Because the flow resistance by buildings is taken account as the drag force in BH model, the ground surface roughness coefficient should be smaller than BR model in which the building drag resistance is conveniently included in the surface

40   roughness. Therefore, we adopted the smaller value for Manning's *n* for the "city center area where BH model was used". However, we agree to reviewer's comment that larger roughness coefficient should be taken for "surrounding areas where we adopted BR model". Therefore, we applied the values of Manning's *n* proposed by Bunya (2010), referring Bricker's paper for the "surrounding area" in the new manuscript. The new results did not show much difference in the "city center area" from

those in the old manuscript. We replaced the new calculation results (Fig.14-19), and added Bunya's work in the text and reference list.

[Revised]

**Page 5 Line 14-16**

Manning's roughness coefficient was assigned as described by Bunya (2010) and Bricker et al. (2015) for each land-use classification, as presented in **Table 1** and as shown with colors in **Fig. 8(b)**.

**Page 10 Line 10-13**

Bunya, S., Deitrich, J. C., Westerink, J. J., Ebersole, B. A., Smith, J. M., Atkinson, J. H., Jensen, R., Resio, D. T., Luettich, R. A., Dawson, C., Cardone, V. J., Cox, A. T., Powell, M. D., Westerink, H. J., and Roberts, H. J.: A High-Resolution Coupled Riverine Flow, Tide, Wind, Wind Wave, and Storm Surge Model for Southern Louisiana and Mississippi. Part I: Model Development and Validation. Monthly Weather Review, 18, 345–377, 2010.

[Comment-6]

P6 L13 if the local resident's video is available (i.e., YouTube), you should cite that reference here.

[Reply-6]

We will add the URL of the website to the reference list.

[Revised]

**Page 7 Line 11-12**

A local resident recorded a video (YouTube 2013) recording of tsunami waves from the point shown as the yellow dot in the direction indicated by the blue arrow presented in **Fig. 14(a)**.

**Page 12 Line 17**

YouTube, URL: http://www.youtube.com/watch?v=aQj2zn5Axmk (Referred on June 2013)

[Comment-7]

P6 L28 The fact that the Kamaishi bay-mouth breakwater was ignored should be justified more, as the breakwater had an effect on delaying tsunami arrival time onshore, and also mitigated flood elevation and speed onshore. See for example, Tomita et al. 2012. Effect of breakwaters on reducing flow depth during the Great East Japan Tsunami. Journal of JSCE, series B2 (Coastal Engineering).68(2):I_156-60.

[Reply-7]

We agree reviewer's comment that calculation condition at the bay mouth was different from the actual situation. But, we hope the reviewer understand that the point of our paper is to consider the effect of dense building arrangement on the tsunami run-up flow. We know Tomita et al. (2012) investigated the effect of breakwater on the tsunami propagation into the bay by comparing "distinctive three calculations"; with the breakwater before tsunami arrival; with damaged breakwater configuration measured after the tsunami; and without breakwater, while they did not show the tsunami wave deformation in the process of breakwater destruction. It is still remained for future study. Because of the uncertainness, we did the elaborate photo image analysis for tsunami wave height just near the coast line in order to examine the calculated time series near the coast line could be used for the run-up calculation in the city center area. We hope again the reviewer understand the point of this study and our efforts.

We added the following sentence at the end of "*3.2.1 Ground surface elevation*" in the new manuscript in order to make clear that the tsunami propagation during the collapse of breakwater is still remained for future study.

[Revised]

**Page 4 Line 32-36**

Tomita et al. (2012) investigated the effect of breakwater on the tsunami propagation into the bay by comparing three calculations; with the breakwater before tsunami arrival; with damaged breakwater configuration measured after the tsunami; and without breakwater, while the actual process of breakwater destruction is still remained for future study. Therefore in this study, the damaged configuration measured after the tsunami (Takahashi et al. 2011) was assumed for calculation.

**Page 12 Line 12-13**

Tomita, T., Yeom, G., Ayugai, M., and Niwa T.: Effect of breakwaters on reducing flow depth during the Great East Japan Tsunami. Journal of JSCE, series B2 (Coastal Engineering). 68(2), I_156-I_160, 2012.

We added the purpose of the photo image analysis at the beginning of section *"4.1.1 Field data analysis"* in the new manuscript in order to make sure our consideration.

[Revised]

**Page 6 Line 3-6**

As described earlier, the breakwater at the bay mouth was considered with damaged configuration measured after the tsunami because of the uncertainty of its destruction process. In this study, therefore, time series of tsunami wave height near the coast line were obtained using image analysis of digital photographs taken by residents. Using them, we examined the calculated time series near the coast line for use in run-up calculations in the city center area.

[Comment-8]

Section 5.3. The protection given to inland buildings due to shielding by concrete buildings near the coast reminds me of a paper I saw by Takagi et al (2015) Assessment of the effectiveness of general breakwaters in reducing tsunami inundation in Ishinomaki. Coastal Engineering Journal. 2014 Dec;56(04):1450018. They may have discussed similar effect.

[Reply-8]

We guess the year of publication by Takagi et al. was "2014" though the reviewer-1 wrote "2015". In our understanding, the main topic of their numerical study using BR model was the tsunami attenuation by breakwater surrounding the port of Ishinomaki. In the same paper, they suggested that the damage of houses was smaller behind a large concrete building "from aerial photograph observation", but it was "not from numerical simulation"; their calculation was based on BR model, which could not estimate the effect of each building footprint.

We will insert the following sentence in 5.3.

[Revise]

**Page 8 Line 21-22**

Some reports have suggested that large buildings protected the houses behind them from tsunami impact (e.g., Matsutomi et al., 2012; Takagi et al., 2014).

**Page 11 Line 19-21**

Matsutomi, H., Yamaguchi, E., Naoe, K., and Harada, K.: Damage Conditions to Reinforced Concrete Buildings and Coastal Black Pine Trees in the 2011 Off Pacific Coast of Tohoku Earthquake Tsunami, J. JSCE, Ser. B2 (Coastal Engineering), 68(2), I_351-I_355, 2012. (in Japanese)

PS. We makeed native check before submitting final revised manuscript.

**Response to Reviewer-2:**

[Reviewer's Comment: Black, Author's Comment: Blue, Author's changes in manuscript : Green]

**GENERAL COMMENTS:**

The topic is suitable for the journal since it addresses an issue which could be of interest to the scientific community. The document is up to the international standards and the length of the paper is adequate. **High-resolution modeling of tsunami run-up flooding: A case study of flooding in Kamaishi City, Japan, induced by the 2011 Tohoku Tsunami** has been analysed with interesting conclusions. The results obtained with the developed numerical model present an interesting replication of the recorded data. However, some more explanations are needed in some chapters, in order make it easier the reading and understanding of the study. In addition, the introduced indicator *Z*, is here discussed.

The reviewer would like to give some comments and suggest corrections in order to increase its overall significance.

[Comment-1]

**Abstract**: Although the use of *U* to represent the flow velocity is quite common and it is explained in the chapter 5.2, the abstract must be standalone and thus, the definition of Hmax and Umax must be given.

[Reply]

We added the definitions of the variables. Following another reviewer (#1), we changed the flow intensity indicator to $(hU^2)_{Max}$, the maximum of momentum flux, and we added the explanation of *h* and *U* in the abstract, too.

[Revised]

**Page 1 Line 18-20**

Spatial mapping of an indicator for run-up flow intensity ($I_F = (hU^2)_{max}$, where *h* and *U* respectively denote the inundation depth and flow velocity during the flood). shows fairly good correlation with the distribution of houses destroyed by flooding.

The presence of the results of numerical simulations (lines 18-20) must be adequately presented. The addition of a sentence like "As a possible mitigation measure, the influence of the buildings in the flowing has been addressed…" would increase the text flow.

[Reply]

Following your suggestion, we revised the sentence about the results. The English was checked by an English native speaker before submitting the final manuscript.

[Revised]

**Page 1 Line 20-23**

As a possible mitigation measure, the influence of the buildings in the flowing has been addressed by a numerical experiment for solid buildings arrayed alternately in two lines along the coast. The results show that the buildings can prevent seawater from flowing straight to the city center while maintaining access to the sea.

[Comment-2]

**1.-Introduction:**

The building array treatments are widely explained. But this wide explanation distract from the objective of the paper. A briefer explanation is suggested since the references are enough to study it if necessary. In addition, and this is something common all along the paper, the structure of the chapters is not clear. The **inclusion of a paragraph explaining what the reader is going to find on each chapter is needed to improve the understanding.** If not, although each part is well explained the reader lose their sense of the bigger picture.

[Reply]

We compacted the description of building array treatments and introduction of existing studies, and added the introduction of chapter-structure at the end. The new introduction was checked by an English native speaker before submitting final manuscript.

In the introduction it is not mentioned that the model has been applied as well to study the influence of the concrete buildings. One of the main points of the study is the application of an alternative mitigation measure (not just a seawall) to reduce the tsunami action and to allow, at the same time, the normal work on marine industries.

[Reply]

We added a sentence about the numerical experiment on the influence of buildings along the coast on tsunami intrusion into the city.

[Revise]

**Page 1 Line 27- Page 2 Line 28**

[revised manuscript text omitted]

[Comment-3]
**3.- Methods and materials:**
An introduction must be included (between 3 and 3.1) to explain to the reader what they are about to find in this chapter.

[Reply]
Following [Comment 4] of reviewer-2, we moved Site Description (Chapter-2 in the old manuscript) to the beginning of Application (Chapter-3 in the new manuscript). The Method and materials (Chapter-3 in the old manuscript) becomes Chapter-2 in the new manuscript. In addition, we changed the chapter title "Methods" because we move 3.2 – 3.4 (old manuscript) to the chapter for validation of calculation results your next suggestion. We inserted a short sentence between 2 and 2.1 as follows:

[Revised]
**Page 2 Line 30-32**
Considering the openings of wooden houses such as doors, windows or cracks and slits caused by tsunami effects, the shallow

water BH model was improved to express the effects of wall permeability by introducing the "assumption of internal hydraulic conditions" on line segments where the walls were located. The seawall overtopping was considered similarly.

The characteristics of the model are well explained and referred. Is this model new or has it been presented before? If it is new it should be said clearly, or even named.

[Reply]
This is a new model. We emphasized our original idea "assumption with internal hydraulic conditions" with double quotation marks in the introduction for Chapter-2 (see the above sentence).

In this chapter the characteristic of the numerical model, the application case data sources, and verification data sources are presented together.. These 3 different parts should be separated in order to make it easier the understanding, because they present independent parts of the study. In addition the verification data and the results can be explained together what would improve the overall understanding. This reviewer suggests the change of the structure of chapters 3 and 4 to:
Chapter 3. The numerical model (including chapters 3.1 and 3.2)
Chapter 4. Application case: Kamaishi port under 2011 event.
Introduction explaining the 2011 event
4.1 Mesh generation (including 3.3.1, 3.3.2, and 3.3.3)
4.2 Calculation condition (including 3.4)
Chapter 5. Validation of the results. Include an introduction explaining that the results of the numerical simulations presented in the previous chapter are here presented and compared to those real data recorded. 3 comparisons:
5.1 Tsunami wave height near the coast (including 3.5.1 and 4.1)
5.2 Local highest water surface (including 3.5.2 and 4.2)
5.3 Wave front propagation on streets (including 3.5.3 and 4.3)

[Reply]
Thank you for your comment. We thought in the first draft as you suggested. We will change the structure of final manuscript as follows:

1. Introduction
2. Method: 2.1 Numerical model, 2.2 Assumption of internal boundary conditions.
3. Application case: 3.1 Site description, 3.2 Model set-up (3.2.1 Topographical conditions, 3.2.2 Seawalls and building footprints, 3.2.3 Mesh generation), 3.3. Hydraulic condition for calculation.
4. Validation of results: 4.1 Tsunami wave height near the coast (4.1.1 Field data analysis, 4.1.2 Verification of results), 4.2. Local highest water surface (4.2.1 Field data sources, 4.2.2 Verification of results), 4.3 Wave front propagation on streets (4.3.1 Field data analysis, 4.3.2 Verification of results).
5. Discussion: 5.1. Permeability constant, 5.2. Tsunami effects on houses, 5.3. Tsunami reduction effects concrete buildings along the coast.
6. Conclusions

Again, each chapter must contain an introduction.

[Reply]
We inserted a short introduction for each chapter. These parts were checked by an English native speaker before submitting the final manuscript.

[Revised]
[Chapter 3: Application case]
**Page 3 Line 29-32**
The 2011 off the Pacific Coast of Tohoku Earthquake, with 9.0 in seismic magnitude, hit the northeastern Pacific coast of Japan on March 11, 2011. The total death toll including missing persons reached about 18,000, 90% of whom were killed by the tsunami which struck soon after the earthquake in low-lying urban areas on the coast. Kamaishi City was one of the severely damaged municipalities.

[Chapter 4: Validation of results]
**Page 5 Line 35-37**
Data of many kinds were collected by academic groups, governments, and municipalities after the earthquake. The results obtained from the numerical simulation described in the previous chapter are presented herein and are compared to those real data.

[Chapter 5: Discussion]
**Page 7 Line 23-26**
The reasonable value of $C$, unknown parameter in the model, is discussed in this chapter based on observed data presented in the previous chapter. Then, the tsunami effects on houses are estimated by introducing an indicator for tsunami run-up

intensity. Furthermore, effects of rigid building arrays along the coast are tested numerically as a possible mitigation measure to reduce the hydraulic impact indicator in the city center.

**5.- Discussion:**

An introduction explaining the 2 aspects that are in this chapter (C and Z) is needed.

[Reply]
We added the introduction as written above.
[Revised]
Page 7 Line 23-26

**5.2.** Here the indicator Z=U max*Hmax is presented. This is the product of the maximum inundation depth and the maximum flow velocity during the flood. However, the maximum water depth and the maximum flow velocity are not always simultaneous. The value that should be considered is *Z=(U*H)max,* which is the real maximum value of the product. The indicator must be recalculated or an explanation is needed to maintain the original expression.
This product is used to estimate the human instability hazard (Jonkman et al., 2008)
Jonkman, S., Vrijling, J., and Vrouwenvelder, A.: Methods for the estimation of loss of life due to floods: a literature review and a proposal for a new method, Nat. Hazards, 46, 353–389, doi:10.1007/s11069-008-9227-5, 2008.

[Reply]
Thank you for the suggestion. As mentioned in the reply for the comment #2 of reviewer-1, we finally adopted $(hU^2)_{Max}$, which is flow momentum flux, for flow intensity indicator. The spatial distribution characteristics of the new indicator was basically same as the old indicator, and the points in discussion are same as before.

**SPECIFIC COMMENTS**

**Page 1 Line 10:** shallow *water* equations
[Reply]
We corrected the mistake in the new manuscript.
[Revised]
**Page 1 Line 11**

**Page 1 Line 39:** The reference Gallinen must be Gallien

[Reply]
We corrected the mistake in the new manuscript.
[Revised]
**Page 2 Line 3**

**Page 2 Line 34:** permeability constant, *C* (from..

[Reply]
This part was eliminated from the introduction.

**Page 6 Line 7:** It is not included in the text the reference of the survey. In the reference chapter it is included the 2011 tohoku earthquake tsunami joint survey, but it must be referred in the text.

[Reply]
We cited their work in the new manuscript and added the website to the reference list.
[Revised]
**Page 6 Line 32**

**Page 6 Line 30:** The influence of the port in the flooding was cited by Tomita in
*T. Tomita, G.-S. Yeom, M. Ayugai, T. Niwa, Breakwater Effects on Tsunami Inundation Reduction in the 2011 off the Pacific Coast of Tohoku Earthquake, J. Japan Soc. Civ. Eng. Ser. B 2(Coastal Eng. 68 (2012) 4–8.*
In view of this a comment on the no-consideration of the port in the simulation, as well as the citation of Tomita´s paper must be included.

[Reply]
We added the following sentence at the beginning of section "*3.2.1 Topographical conditions"*, and cited Tomita's paper in reference list.

[Revised]:
**Page 4 Line 32-36**

Tomita et al. (2012) investigated the effect of breakwater on the tsunami propagation into the bay by comparing three calculations: with the breakwater before tsunami arrival, with damaged breakwater configuration measured after the tsunami, and without breakwater, whereas the actual process of breakwater destruction remains as a subject for future study. Therefore in this study, the damaged configuration measured after the tsunami (Takahashi et al. 2011) was assumed for calculation.

We also added the following sentence at the beginning of section *"4.1.1 Field data analysis"* to express our consideration about the breakwater destruction.
[Revised]
**Page 6 Line 3-6**

As described earlier, the breakwater at the bay mouth was considered with damaged configuration measured after the tsunami because of the uncertainty of its destruction process. In this study, therefore, time series of tsunami wave height near the coast line were obtained using image analysis of digital photographs taken by residents. Using them, we examined the calculated time series near the coast line for use in run-up calculations in the city center area.

**Page 7 Line 10:** Is this video available on the internet? If so, a reference would be interested.

[Reply]
We added the URL of the website to the reference list.
[Revised]
**Page 7 Line 11-12**

A local resident recorded a video (YouTube 2013) recording of tsunami waves from the point shown as the yellow dot in the

direction indicated by the blue arrow presented in **Fig. 14(a)**.

**Page 12 Line 17**

YouTube, URL: http://www.youtube.com/watch?v=aQj2zn5Axmk (Referred on June 2013)

**Page 8 Line 1:** The expression includes hmax, but in the rest of the manuscript it is called Hmax.

[Reply]
Equation (3) was changed to $(hU^2)_{Max}$, as mentioned before, and we writed correctly.
[Revised]

**Page 8 Line 13**

Equation (3)

**FIGURES**:
Figure 11 is called for the first time in page 6 line10, but the symbols contained in it are not explained until Figure 15 is called in line 34. They should be explained in the foot of the figure.

[Reply]
We added the explanation of the symbols in the foot of the figure.
[Revised]
**Page 17 Figure 12**

Figure 14a. In this figure are depicted the water levels at 4 points, but just the results of the model for the P3 are represented. However there are just 3 points photographed in P3. Other points have many more dots so it seems logical to depict other point time series instead of P3. In addition, the fact that all the dots (even those from other points like P1, P2 and P4) agreed fairly well in the P3 time series is important as to be highlighted.

[Reply]
Calculated water levels at the four points were almost same because they were very close to one another. Therefore, we showed the calculation result at P3 which was at the center. We mentioned it in the new manuscript.
[Revised]
**Page 6 Line 18-19**
The *P3* located at the center of measurement area was selected for plotting of the calculation results because the four points were mutually very close and calculation results were almost identical.